# Bayesian inversion of a CRN depth profile to infer Quaternary erosion of the northwestern Campine Plateau (NE Belgium)

Eric Laloy[1], Koen Beerten[1], Veerle Vanacker[2], Marcus Christl[3], Bart Rogiers[1], Laurent Wouters[4]

[1]Institute Environment-Health-Safety, Belgian Nuclear Research Centre (SCK•CEN), Mol, 2400, Belgium

[2]Georges Lemaitre Centre for Earth and Climate Research, Earth and Life Institute, Université catholique de Louvain, Louvain-la-Neuve, 1348, Belgium

[3]Laboratory of Ion Beam Physics, ETH-Zurich, Zurich, 8093, Switzerland

[4]Long-term RD&D department, Belgian National Agency for Radioactive Waste and enriched Fissile Material (ONDRAF/NIRAS), Brussels, 1210, Belgium

*Correspondence to*: Koen Beerten (kbeerten@sckcen.be)

**Abstract.** The rate at which low-lying sandy areas in temperate regions, such as the Campine plateau (NE Belgium), have been eroding during the Quaternary is a matter of debate. Current knowledge on the average pace of landscape evolution in the Campine area is largely based on geological inferences and modern analogies. We performed a Bayesian inversion of an in-situ produced $^{10}$Be concentration depth profile to infer the average long-term erosion rate together with two other parameters: the surface exposure age and the inherited $^{10}$Be concentration. Compared to the latest advances in probabilistic inversion of cosmogenic radionuclide (CRN) data, our approach has the following two innovative components: it (1) uses Markov chain Monte Carlo (MCMC) sampling and (2) accounts (under certain assumptions) for the contribution of model errors to posterior uncertainty. To investigate to what extent our approach differs from the state-of-the-art in practice, a comparison against the Bayesian inversion method implemented in the CRONUScalc program is made. Both approaches identify similar maximum a posteriori (MAP) parameter values, but posterior parameter and predictive uncertainty derived by the method taken in CRONUScalc is moderately underestimated. A simple way for producing more consistent uncertainty estimates with the CRONUScalc-like method in the presence of model errors is therefore suggested. Our inferred erosion rate of 39 ± 8 mm/1$\sigma$) is relatively large in comparison with landforms that erode under comparable (palaeo-)climates elsewhere in the world. We evaluate this value in the light of the erodibility of the substrate and sudden base level lowering during the Middle Pleistocene. A denser sampling scheme of a two-nuclide concentration depth profile would allow to better resolve the inferred erosion rate, and to include more uncertain parameters in the MCMC inversion.

## 1 Introduction

The Campine area is a sandy region which covers part of northeastern Belgium and the southern Netherlands (Fig. 1). It is part of the European sand belt and is drained by rivers that belong to the Scheldt basin. The Campine area roughly coincides with

the geological Campine Basin, being the southeastern part of the North Sea Basin. From a geodynamic point of view, the Campine Basin is located in an intermediate position in between the rapidly subsiding Roer Valley Graben in the north, and the uplifting Brabant and Ardennes Massifs in the south (Fig. 2). The Campine Basin has witnessed a long Cenozoic burial history. Post-Rupelian marine and estuarine deposition during the last 30 Myr almost exclusively consists of (glauconite-rich) sand, up to 300 m thick (Vandenberghe et al., 2004). From the Early to Middle Pleistocene onwards, terrestrial conditions become dominant with deposition of a thick series of fluvial sand and gravel from the rivers Meuse and Rhine (Figs. 2 and 3). In contrast to what the basinal setting of the Campine region would suggest, distinctive topographic features are preserved in the landscape. An illustrative example is the Campine Plateau, which shows a topographic relief of ca. 50 m relative to the surrounding areas (Fig. 3). To date, quantitative data on the amount and rate of Quaternary erosion of the Campine landscape and the Scheldt basin in general are missing. This stands in contrast to the availability of long-term erosion data from e.g. in-situ produced cosmogenic nuclides for the Meuse and Rhine basins (e.g., Schaller et al., 2001; Dehnert et al., 2011; Rixhon et al., 2011). Such data on catchment-wide erosion rates at multi-millennial timescales are crucial to determine background geological erosion rates to evaluate anthropogenic morphodynamics (Vanacker et al., 2007a), to provide calibration data for landscape evolution models (Bogaart and van Balen, 2000; Foster et al., 2015; Campforts et al., 2016), and to assess the overall stability of the landscape in the framework of long-term management of radioactive waste (Van Geet et al., 2012).

Cosmogenic radionuclides (CRN's) have proven useful to quantify geomorphological processes over timespans covering the last 2 challer et al., 2001). Geomorphological surfaces can be dated by measuring the concentration of in-situ produced cosmogenic nuclides (e.g., $^{10}$Be and $^{26}$Al) that accumulated at the Earth's surface (Dunai, 2010; Hancock et al., 1999). As the observed cosmogenic nuclide concentration of a given outcrop is a function of its exposure age and denudation rate, stable (i.e. non-eroding) landforms provide optimal sampling locations for exposure dating (e.g., Rixhon et al., 2011). Most landforms are subject to erosion during exposure, resulting in a decrease of the cosmogenic nuclide concentrations with increasing surface denudation rate (e.g., Dehnert et al., 2011). Braucher et al. (2009) showed that the exposure age (and post-depositional denudation rate) of eroding landforms can be constrained based on a deep (> 1.5 m) depth profile of a single cosmogenic nuclide that is sampled at regular intervals.

The accumulation of in-situ produced cosmogenic nuclides in eroding surfaces is a mathematical function with three parameters that are typically unknown a priori: the post-depositional denudation rate, $E$ [m/Myr], the exposure age, $t$ [yr], and the inherited concentration or inheritance, $N_{inh}$, [atoms/g]. Unknown model parameters can be estimated by inverse modeling of CRN concentration vs. depth profiles. In this procedure, one iteratively proposes new parameter values until the model fits the observed data up to a given precision. This has been done for estimating $E$ and $t$ by, e.g., Siame et al. (2004) and Braucher et al. (2009). Yet, model and measurement errors together with (measurement) data scarcity introduce considerable uncertainty in the optimized model parameters. The method by Braucher et al. (2009) accounts to some extent for analytical measurement errors, as it generates several CRN concentration profiles consistent with the (analytical) measurement errors, computes for

each model parameter set (e.g., a *t-E* pair) within a grid search the corresponding data misfits and retains the median misfit as the performance associated with a given parameter set. This allows for deriving a robust unique solution but does not quantify model parameter uncertainty and ignores model errors (that is, the model is assumed to be perfect). To assess model parameter uncertainty, Hidy et al. (2010) proposed to perform plain Monte Carlo (MC) sampling from the pre-specified prior parameter distributions, rank the resulting solutions according to fitting performance and retain a certain percentage (typically 5%) of the best performing solutions to compute parameter uncertainty estimates.

A more comprehensive quantification of parameter and prediction uncertainty is provided by the Bayesian framework. This approach uses Bayes theorem to represent parameter uncertainty by a multivariate "posterior" probability distribution. The latter is given by the (normalized) product of a "prior" probability distribution, which represents available prior information, with a "likelihood" function, that encodes the deviations of the simulated (CRN) concentration data from the measured ones. Providing that the assumptions underlying the likelihood model are met, the posterior probability density function (pdf) contains all necessary information about the inferred parameters. Marrero et al. (2016) implemented Bayes rule into the CRONUScalc program to derive the posterior parameter pdf. The approach taken by Marrero et al. (2016) is based on a MC variant where sampling is performed over a regular, 3-dimensional lattice covering the prior ranges for $E$, $t$ and $N_{inh}$. It therefore requires a sufficient grid resolution to minimize the risk of missing $E$ - $t$ - $N_{inh}$ combinations with substantial posterior density. More importantly, the formulation by Marrero et al. (2016) considers solely the CRN measurement error(s) as source of uncertainty. This is theoretically valid only if the CRN model can fit the measurement data within the measurement error(s). In this work, we derive the posterior parameter pdf by state-of-the-art Markov chain Monte Carlo (MCMC) simulation (see, e.g., Robert and Casella, 2004), accounting not only for measurement errors but also (under certain assumptions) for model errors. Furthermore, we compare our Bayesian inversion approach with that of Marrero et al. (2016), illustrate the similarities and differences between the two approaches, and propose a simple fix for making the uncertainty estimate from Marrero et al. (2016) more consistent in the presence of model errors.

The overall objective of this study is to infer within a Bayesian framework the potential post-depositional denudation of the northwestern Campine Plateau. This part of the Campine Plateau is drained by the Kleine Nete river, which belongs to the larger Scheldt basin. It is an interesting test case because the northwestern edge of the Campine Plateau is covered by coarse gravelly unconsolidated sand from the Early-Middle Pleistocene Rhine and thus constitutes a fluvial terrace for which the depositional age nor the exposure age is well constrained (Beerten et al., in press).

## 2 Geomorphological evolution of the Campine area

The post-marine hydrographical evolution of the Campine area started with the final retreat of the sea during the Neogene, as a result of systematic sea level lowering and overall uplift of the bordering areas around the southern North Sea (Miller et al.

2005; Cloething et al. 2007). During the Early Pleistocene, the Meuse followed an eastern course from Liège to the region north of Aachen where it merged with the Rhine (Fig. 2). Tectonic movements along Roer Valley Graben faults, and uplift of
the northern margins of the Ardennes-Eifel massif caused the Meuse to breach through its northern interfluve and to follow a completely different course. At the same time, the Rhine shifted its course as well, flowing into the northern part of the Campine
area where it merged with the Meuse (Fig. 2). Age control is limited, but this event probably took place around 1 Ma at the earliest, since the confluence area of both rivers was situated in the southeastern part of the Roer Valley Graben prior to 1 Ma,
and the area that covers the Campine Plateau today was drained by local 'Belgian' rivers until that time (Westerhoff et al., 2008). Both rivers shifted their course towards a more eastern position by 0.5 Ma at the latest, given the absence of Rhine
deposits younger than 0.5 Ma in the depocenter of the Roer Valley Graben (Schokker et al. 2005). The deposits that cover the Campine Plateau are often correlated with the upstream Main Terraces of the Meuse and High Terraces of the Rhine (Paulissen,
1973). Westaway (2001) provides a time window for deposition of Rhine sediments west of the Ville Ridge (High Terraces HT2 and HT3) between 0.5 Ma and 1 Ma. The Rhine sediments on top of the Campine Plateau have been attributed to the
Sterksel Formation, which was deposited between ca. 0.6 Ma and 1.1 Ma according to van Balen et al. (2000), and between ca. 0.75 Ma and 1 Ma (post-Jaramillo Early-Pleistocene) according to Gullentops et al., (2001). Recently, deposits from the
High Terraces of the Rhine between Bonn (Germany) and Venlo (the Netherlands) have been dated to $750 \pm 250$ ka and $740 \pm 210$ ka using in-situ produced cosmogenic radionuclides (Dehnert et al., 2011). Similarly, Meuse terrace deposits in the
Liège area (Romont, Belgium) that are generally assumed to correspond with the series of Main Terrace deposits, have been dated to $725 \pm 120$ ka using the same technique (Rixhon et al., 2011).

During the Middle Pleistocene, the hydrography of northern Belgium drastically changed due to the 'opening' of the English
Channel (Vandenberghe and De Smedt 1979; Fig. 2). Various studies (Gibbard 2007; Gupta et al. 2007; Toucanne et al. 2009) link the opening of the English Channel to the catastrophic drainage of a large proglacial lake during marine isotope stage 12
(MIS 12), approximately 450 ka ago (Elsterian). The 450 ka event triggered the formation of a buried palaeo-channel system known as the Flemish Valley, with extensions towards the south and the east (Tavernier and De Moor, 1974). The Nete
catchment is generally considered as the eastern extension of the Flemish Valley. At present, the Campine Plateau is a landform that markedly stands out with respect to its surroundings. It is a fluvial terrace covered by coarse gravelly Meuse deposits in
the south(east) and sandy Rhine deposits in the north (Fig. 3). The sediments have proven to feature a periglacial palaeoenvironment and were deposited by braided rivers (Paulissen, 1973 and 1983). The Campine Plateau can be considered
as a classical case of relief inversion, given its prominent position in the landscape (Paulissen, 1983; Fig. 4). Undoubtedly, the area west of the Campine Plateau experienced prolonged phases of erosion and denudation after the Rhine had left the region,
around 0.5 Ma at the latest (Fig. 4b; Beerten et al., in press).

## 3 Material and methods

### 3.1 Cosmogenic radionuclide profiling

Cosmogenic radionuclides (CRN) allow us to quantify geomorphological processes over timespans covering the last 1 Myr. In this study, we use the concentration vs. depth profile of a single in-situ produced CRN ([10]Be) to constrain the post-depositional denudation rate, $E$ [m/Myr] of the fluvial terrace. The accumulation of CRN, $N_{total}(z,t)$ [atoms/g], in an eroding surface can be described by a mathematical function composed of two terms that represent the inherited CRN concentration of the fluvial sediment, $N_{inh}$ [atoms/g]., and the post-depositional production of CRN, $N_{exp}(z)$:

$$N_{total}(z,t) = N_{inh} + \sum_i \frac{P_i(z)}{\lambda + \frac{\rho E}{\Lambda_i}} e^{-(\rho(z_0 - Et)/\Lambda_i)} \left(1 - e^{-\left(\lambda + \frac{\rho E}{\Lambda_i}\right)t}\right) \tag{1}$$

where $E$ is expressed in cm/yr (m/Myr $\times$ 10$^{-4}$), $t$ [yr] is the exposure age, $\lambda$ [1/yr] the decay constant ($\lambda$ = ln 2 /$t_{1/2}$), $z_0$ the initial shielding depth ($z_0 = E \times t$), $\rho$ [g/cm³] the density of the overlying material, and $\Lambda_i$ [g/cm²] the attenuation length. The production rate of CRN, $P_i(z)$ [atoms/g/yr], is a function of the depth, $z$ [cm], below the surface as:

$$P_i(z) = P_i(0) e^{-\frac{z\rho}{\Lambda_i}} * \tag{2}$$

The subscript 'i' indicates the different production pathways of in-situ produced [10]Be via spallation, muon capture and fast muons following Dunai (2010). In this study, the relative spallogenic and muogenic production rates are based on the empirical muogenic to spallogenic production ratios established by Braucher et al. (2011), using a fast muon relative production rate at SLHL of 0.87% and slow muon relative production rate at SLHL of 0.27%. The effective attenuation length is here equal to the apparent attenuation length as the depth profile was taken on a horizontal surface. The effective attenuation length for the sampling position was obtained using Table 4 in Marrero et al. (2015), and equals 152 g/cm². For fast and stopped muons, the attenuation length was set at resp. 1500 and 4320 g/cm² following Braucher et al. (2011). Production rates were scaled following Stone (2000) with a sea level high latitude production rate of (4.25 ± 0.18) atoms/g/yr (Martin et al., 2017). The latter represents the regionally averaged SLHL production rate for Europe. The bulk density, $\rho$, of the studied fluvial sediment was set to 1.7 g/cm³, which is consistent with the average value of upper Neogene and Quaternary sediments in the region (Beerten et al., 2010). A half-life of (1.387 ± 0.012) $\times$ 10$^6$ y was used for [10]Be following Cmeleff et al. (2010). CosmoCalc add-in for Excel was used to calculate the scaling factors. Given the flat topography of the Campine Plateau, topographic shielding was negligible and therefore not corrected for (Norton and Vanacker, 2009).

## 3.2 Sampling and analytical methods

The depth profile was sampled in a sand pit (SRC-Sibelco NV) on the northwestern edge of the Campine Plateau (Fig. 4a and b). The altitude of the sampling spot is ca. 47 m (Tweede Algemene Waterpassing), while the crest of the plateau further east
reaches an altitude of ca. 48 m. The almost 4 m thick sequence is composed of medium-grained quartz-rich fluvial sands, overlain by a thin layer (35 cm thick) of fine-grained aeolian sand (Fig. 5). Detailed grain-size characteristics of the fluvial
sand are given in Fig. 6. Note that sample depth is given with reference to the top of the fluvial sands. The lowermost unit A consists of medium sand with mode and median in the range between 250-500 µm, while a significant portion of grains coarser
than 500 µm is present. Unit B is finer with a median grain size of ca. 250 µm and virtually no coarse sand (i.e., > 500 µm). Unit C consists of coarse sand (median grain size more than 500 µm) with a significant amount of fine gravel fragments. The
next unit (E) is the finest unit of the sequence, with mode and median below 250 µm. Sediments from unit F are generally finer than those of units A and C, but coarser than those from units B and D. Mode and median are in the range between 250-
500 µm. Finally, unit G represents a thin layer of fine sand, interpreted as Late Pleistocene aeolian deposits. Note that samples MHR-II-06 and MHR-II-04 are taken in much finer sand beds compared to the other samples.
From the depth profile, ten samples were collected for CRN analysis at depths ranging from 10 to 320 cm below the fluvial-aeolian contact, from which 9 were analyzed. Samples were more or less evenly spread out over the sequence, although the
sampling density was higher towards the top (Table 1). Samples were taken as bulk samples of 1.5 kg, over a depth interval of 10 cm. Samples were sieved, and the 500-1000 µm grain size fraction was used for sample preparation, except for the fine-
grained sand samples MHR-II-04 and MHR-II-06 where the 250-500 µm fraction had to be used.

    Samples were prepared at the University of Louvain Cosmogenic Isotope Laboratory (Louvain-la-Neuve). In-situ produced

$^{10}$Be was extracted from purified quartz using standard separation methods described in von Blanckenburg et al. (1996) and Vanacker et al. (2007b). Two blanks were processed with the nine samples. Approximately 200 µg of $^{9}$Be carrier was added
to blanks and samples containing 30 to 35 g pure quartz. The $^{10}$Be/$^{9}$Be ratios were measured in BeO targets with accelerator mass spectrometry on the 0.6 MV Tandy at ETH Zurich (Kubik and Christl, 2010). The ratios were normalized to the ETH in-
house secondary standard S2007N with a nominal value of $^{10}$Be/$^{9}$Be of $28.1 \times 10^{-12}$ (Kubik and Christl, 2010) which is in agreement with a half-life of 1.387 Myr (Chmeleff et al., 2010). Samples are corrected for the number of $^{10}$Be atoms in their
associated blanks. The analytical uncertainties on the $^{10}$Be/$^{9}$Be ratios of sample and blank are then propagated into the 1σ analytical uncertainty for nuclide concentrations.

## 3.3 Bayesian inference

**3.3.1 Inverse problem**

    To acknowledge that measurements and modelling errors are inevitable, the inverse problem is commonly represented by the

stochastic relationship given by
$\mathbf{d} = F(\mathbf{x}) + \mathbf{e}$                                                                    (3)
where $\mathbf{d} = (1, \dots, N) \in \mathbb{R}^N, N \geq 1$ is the measurement data, $F$ is a deterministic forward model with parameters, $\mathbf{x}$, and the noise term, $\mathbf{e}$, lumps measurement and model errors.
Inversions were performed within a Bayesian framework, which treats the unknown model parameters $\mathbf{x}$ as random variables with posterior probability density function (pdf), $p(\mathbf{x}|\mathbf{d})$, given by

$$p(\mathbf{x}|\mathbf{d}) = \frac{p(\mathbf{d}|\mathbf{x})p(\mathbf{x})}{p(\mathbf{d})} \propto L(\mathbf{x}|\mathbf{d})p(\mathbf{x})$$     (4)


where $p(\mathbf{x})$ denotes the prior distribution of $\mathbf{x}$ and $L(\mathbf{x}|\mathbf{d}) \equiv p(\mathbf{d}|\mathbf{x})$ signifies the likelihood function of $\mathbf{x}$. The normalization
factor $p(\mathbf{d}) = \int p(\mathbf{d}|\mathbf{x})p(\mathbf{x})d\mathbf{x}$ is obtained from numerical integration over the parameter space so that $p(\mathbf{x}|\mathbf{d})$ scales to unity. The quantity $p(\mathbf{d})$ is not required for parameter inference. Unless stated otherwise, in the remainder of this study we will focus
on the unnormalized posterior $p(\mathbf{x}|\mathbf{d}) \propto L(\mathbf{x}|\mathbf{d})p(\mathbf{x})$.

If we assume the residual errors, $\mathbf{e}$, to be normally distributed, uncorrelated and with unknown constant variance, $\sigma_e^2$, the log-
likelihood function can be written as

$e_i = d_i - F_i(\mathbf{x})$                                                                    (5)
$L(\mathbf{x}|\mathbf{d}) = \frac{1}{\sqrt{2\pi}\sigma_e} \exp\left[\frac{-1}{2\sigma_e^2} \sum_{i=1}^{N} e_i^2\right]$     (6)
For numerical stability, it is however often preferable to work with the log-likelihood function, $\ell(\mathbf{x}|\mathbf{d})$, instead of $L(\mathbf{x}|\mathbf{d})$
$\ell(\mathbf{x}|\mathbf{d}) = -\frac{N}{2}\ln(2\pi) - N\ln(\sigma_e) - \frac{1}{2\sigma_e^2}\sum_{i=1}^{N} e_i^2$     (7)
The variance of the residual errors, $\sigma_e^2$, can be fixed beforehand or sampled jointly with the other model parameters $\mathbf{x}$. Note that by fixing $\sigma_e^2$ to a known measurement error, $\sigma_m^2$, one implicitly assumes that the model is able to describe the observed
system up to the observation errors. This might not be realistic in environmental modelling, where models are always fairly simplified descriptions of a more complex reality. In this work, we therefore jointly infer $\sigma_e$ with $\mathbf{x}$. This accounts for both
measurement and model errors, under the assumption that both types of errors obey a zero-mean uncorrelated and homoscedastic normal distribution.

### 3.3.2 Markov chain Monte Carlo sampling

The inference seeks to estimate the posterior parameter distribution of the model parameters, $p(\mathbf{x}|\mathbf{d})$. As an exact analytical solution for $p(\mathbf{x}|\mathbf{d})$ is not available, we resort to Markov chain Monte Carlo (MCMC) simulation to generate samples from this distribution. The basis of this technique is a Markov chain that generates a random walk through the search space and iteratively finds parameter sets with stable frequencies stemming from the posterior pdf of the model parameters (see, e.g., Robert and Casella, 2004, for a comprehensive overview of MCMC simulation). In practice, the MCMC sampling efficiency strongly depends on the assumed proposal distribution used to generate transitions in the Markov chain. In this work, the state-of-the art DREAM$_{(ZS)}$ (ter Braak and Vrugt, 2008; Vrugt et al., 2009; Laloy and Vrugt, 2012) algorithm is used to generate posterior samples. A detailed description of this sampling scheme including convergence proof can be found in the cited literature and is thus not reproduced herein. Note that the considered CRN data inversion is a fairly simple problem (the model in Eqs. (1-2) is quick and well behaved whereas both the parameter and measurement data spaces are rather low-dimensional). The use of DREAM$_{(ZS)}$ will become even more attractive when considering larger parameter dimensionality.

### 3.3.3 Prior distribution

The prior pdf is a key element of Bayesian inference. This distribution encodes the available prior information about the parameters and balances the effect of the likelihood function on the posterior pdf (Eq. (4)). We assumed the individual prior parameter pdfs to be independent

$$p(\mathbf{x}) = \prod_{i=1}^{N_p} p(x_i) \qquad\qquad (8)$$

with $N_p = 4$ the dimension of $\mathbf{x}$.

Based on the current geological knowledge of the region, we specified a truncated Gaussian prior distribution for $E$, with mean of 30 m/Myr, standard deviation of 30 m/Myr and range of [0, 60] m/Myr. The upper bound of 60 m/Myr is based on (1) geomorphological evidence presented in Fig. 4b, using the altitude difference between the Campine Plateau and the adjacent Kleine Nete floodplain, and (2) the youngest possible age for the Rhine sediments. The lower bound of 0 m/Myr corresponds to the scenario where the Campine Plateau is a residual relief due to erosion-resistance of the covering sediments. Overall, the resulting $p(E)$ is sufficiently vague to avoid over-constraining the inversion while nevertheless discouraging the search to pick up values close to the boundaries that are considered to be (a priori) less likely. A uniform prior distribution in the range [0, 1] Myr was selected for $t$. This is based on the presumed burial age of the Rhine sands covering the Campine Plateau (between 0.5 Ma and 1 Ma) together with geologic evidence on the evolution of the Scheldt basin and more in particular the Nete catchment after the opening of the English Channel (0.45 Ma, see Section 2). In addition, we put a uniform prior pdf with range [1, 35] m on the product $E \times t$. This limits the total erosion that can possibly be inferred from the measurement data to

35 m, as we expect that 35 m of total erosion on top of the Campine Plateau is an absolute maximum. When adding the thickness of Rhine sediment that are covering the top of the Campine Plateau (i.e. 5 to 10 m) to the maximum total erosion,
we obtain a maximum initial thickness of 40 m to 50 m, which corresponds to the thickness of Rhine deposits that are preserved in the deepest part of the Roer Valley Graben (Beerten, 2006; Deckers et al., 2014). The minimum amount of total erosion is
set to 1 m, given the altitude of the sampling position which is slightly (ca. 1 m) lower than the crest of the plateau. The prior distribution for the inherited $^{10}$Be concentration, $N_{inh}$, was assumed to be uniform. This is because solutions for $N(z,t)$ in Eq.
(1) always converge to $N_{inh}$ as $z$ tends to infinity while no $N_{inh}$ measurements are available at z larger than 3.5 m. The $N_{inh}$ parameter was therefore allowed to vary uniformly between $1 \times 10^4$ atoms/g and $9 \times 10^4$ atoms/g. The upper bound of $9 \times 10^4$
at/g is consistent with the highest concentration measured in the profile (Table 2; the inherited concentration cannot be higher than this value). The lower bound was somewhat arbitrarily set at $1 \times 10^3$ at/g, given the fact that zero inheritance is considered
to be very unlikely. Lastly, a so-called Jeffreys (1946) prior of the form $p(\sigma_e) \propto 1/\sigma_e$ was used for $\sigma_e$. This classical choice basically means that one wants to achieve $\sigma_e$ values that are both as small as possible and large enough to be consistent with
the data misfit.
**3.3.4 Comparison with the Bayesian approach in CRONUScalc**

For comparison, we implemented the approach taken by Marrero et al. (2016) in CRONUScalc (CR). For brevity, in the

reminder of this paper we will refer to this method as CRB for "CRONUScalc Bayes". Similarly as our approach, CRB seeks to estimate $p(\mathbf{x}|\mathbf{d})$ using Eq. (4). However, CRB does not generate a set of samples with frequencies stemming from $p(\mathbf{x}|\mathbf{d})$.
Instead it samples the prior pdf, $p(\mathbf{x})$, over a high-resolution 3-dimensional evenly-spaced lattice and computes the (normalized) posterior pdf, $p(\mathbf{x}|\mathbf{d}) = L(\mathbf{x}|\mathbf{d})p(\mathbf{x})/p(\mathbf{d})$, for each grid point. This requires evaluation of $p(\mathbf{d}) =$
$\int L(\mathbf{x}|\mathbf{d})p(\mathbf{x})d\mathbf{x}$, which is done by a trapezoidal integration scheme.

The need to evaluate $p(\mathbf{d})$ by trapezoidal integration implies that CRB cannot use $\ell(\mathbf{x}|\mathbf{d})$ but requires using $L(\mathbf{x}|\mathbf{d})$. The

$L(\mathbf{x}|\mathbf{d})$ formulation taken by CRB is similar as Eq. (5), except for two aspects. First CRB includes the more general heteroscedastic case as well. In other words, the residual errors, $\mathbf{e} = [e_1, ..., e_N]$, can have different variances, $\sigma_{e_1}^2, ..., \sigma_{e_N}^2$.
Second and most important, CRB sets the $\sigma_{e_i}$ to analytical measurement errors, $\sigma_{m_i}$. Using similar notations as in Marrero et al. (2016), CRB uses
$$\chi^2 = \sum_{i=1}^{N} \left( \frac{e_i}{\sigma_{m_i}} \right)^2 \tag{9}$$

$$L(\mathbf{x}|\mathbf{d}) = \prod_{i=1}^{N} \left( \frac{1}{\sqrt{2\pi}\sigma_{m_i}} \right) \exp\left[ -\frac{\chi^2}{2} \right] \tag{10}$$

where $\chi^2$ is a weighted sum of squared residuals (WSSR) also referred to as the chi-square statistic. In this study $\sigma_m$ is assumed to be constant: $\sigma_{m_1}, ..., \sigma_{m_N} = \sigma_m$. Eq. (10) therefore reduces to Eq. (6) but with the standard deviation of the residual errors,
$\sigma_e$, fixed to $\sigma_m$. Thus CRB considers 3 parameters: $E$, $t$ and $N_{\text{inh}}$. With respect to the associated 3-dimensional prior distribution, $p(\mathbf{x})$, we used the assumptions as for our approach (see section 3.3.3).
As stated earlier, no matter whether ones uses a constant $\sigma_m$ or a different $\sigma_{m_i}$ for each residual, $e_i$, fixing the standard deviation(s) of the residual errors to the standard deviation(s) of the measurement errors implicitly assumes that the model
can fit the concentration data within the standard deviation(s) of the measurement errors. If this assumption is not met, then the resulting $p(\mathbf{x}|\mathbf{d})$ estimation will be biased towards underestimation of uncertainty. For the case of constant $\sigma_e$ and $\sigma_m$ in
Equations (6) and (10), respectively, the solution that maximizes $p(\mathbf{x}|\mathbf{d})$, or maximum a posterior solution (MAP), should have a root-mean-square error, $RMSE = \sqrt{N^{-1} \sum_{i=1}^{N} e_i^2}$, that is close to $\sigma_e$ (Eq. (10)) or $\sigma_m$ (Eq. (6)). Otherwise, the chosen
likelihood model is not consistent with the actual data.

### 3.3.5 Predictive uncertainty intervals

A 95% uncertainty interval for the simulated CRN concentrations can be calculated by drawing parameter sets, $\mathbf{x}$, from $p(\mathbf{x}|\mathbf{d})$ and removing the 2.5% largest and lowest values from the associated set of $F(\mathbf{x})$ responses. If all prior assumptions about the
residual error distribution are met, then this 95 % predictive uncertainty interval should encompass 95% of the measurement data.

### 4 Results

### 4.1 CRN measurement data

In general, there is a clear decrease in [10]Be concentration with depth, except for two samples (-04 and -06) which contain higher CRN concentrations (Table 1 and Fig. 6). It is striking that the CRN concentrations are consistently higher for the two
samples were the finer (250-500 µm) grain size fraction was analyzed. Grain size-dependent [10]Be concentrations can point to differences in geomorphological process rates in the regions of sediment provenance as suggested by Carretier et al. (2015).
Alternatively, the negative relation between in-situ produced CRN and grain size might also result from non-stationary sedimentation rates, where samples from the fine-grained layers accumulated CRN during the final stage of the sedimentation
cycle prior to a phase of non-deposition and/or steady state. Apart from the -04 and -06 samples, the CRN concentrations decrease non-linearly with depth, from $(1.5 \pm 0.02) \times 10^5$ atoms/g at 10 cm to $(9.0 \pm 0.2) \times 10^4$ atoms/g at 320 cm (Table 1
and Fig. 6). Because they are not compatible with the other profile data, samples -04 and -06 were excluded from the inversion, thereby leading to a measurement data set of 7 concentrations. These measured concentrations are associated with analytical
measurement errors, $\sigma_{m_1}, ..., \sigma_{m_7}$, in the range of $6 \times 10^3$ atoms/g $- 7 \times 10^3$ atoms/g.

    If we consider the end-member where erosion rates of the Campine Plateau are very low ($E \approx 0$ m/Myr), as one could assume

from its geomorphic setting as an inverted topography, the difference between $N_{\text{total}}(z)$ and $N_{\text{inh}}$ gives the $N_{\text{exp}}(z)$ or the concentration of cosmogenic nuclides that is produced at depth $z$ after deposition of the Rhine sands. The apparent exposure
age of the surface, $t$, can then be reconstructed following Eq. (1). By doing so, we obtain an apparent exposure age of 21.5 ±

1.5 ka, which is in strong contradiction with chronostratigraphical age estimates of the fluvial deposits that cover the Campine

Plateau that range between 0.5 Ma and 1 Ma (see Section 2 and 3.3.2). We advocate that post-depositional erosion has strongly

altered the $^{10}$Be signature of the upper layers of the Rhine sediments at the study site.

**4.2 Inversion**

**4.2.1 Posterior distribution derived by our proposed approach**

We ran DREAM$_{(ZS)}$ for a (serial) total of $150 \times 10^3$ model evaluations. The marginal posterior distributions of the 4 sampled

parameters (including the standard deviation of the residual errors, $\sigma_e$) are depicted in Fig. 7 while bivariate posterior scatter

plots together with iso-density contour lines are presented in Fig. 8. The erosion rate, $E$, is relatively well resolved with a clear

mode around 39 m/Myr (Fig. 7a and 8a-b). The posterior $E$ uncertainty remains however large (Fig. 7a and 8ab), with a

standard deviation (1σ) of 8 m/Myr and a 95% uncertainty interval of [25.8, 57.5] m/Myr. Furthermore, posterior $E$ and $N_{inh}$

values are positively correlated (Table 2 and Fig. 8b) with a linear correlation coefficient of 0.57. The $t$ posterior distribution

shows a weakly expressed mode between ca. 50 ka and 200 ka (Fig. 7b, 8a and 8c) but nevertheless resembles the $t$ prior

density (Fig. 7b) rather closely, except for $t > 0.5$ Myr. Also $t$ does not correlate with other sampled parameters, except for for

$N_{inh}$ (Table 2 and Fig. 8c) with which a linear correlation of -0.38 is observed (Table 2). The $t$ parameter is therefore left largely

unresolved by the inversion. The $N_{inh}$ parameter shows a clear mode around $8.4 \times 10^4$ atoms/g (Fig. 7c, 8b and 8c), which is

lower than the lowest measured value in the profile of about $9.09 \times 10^4$ atoms/g (Table 1). As mentioned already, a moderately

large dependence with $E$ is observed (Fig. 8b and Table 2). The $\sigma_e$ parameter shows an approximately log-normal marginal

posterior with a clear mode around $1 \times 10^4$ atoms/g (Fig. 7d). This is consistent with the achieved RMSE values between

measured and simulated $^{10}$Be. Indeed, the MAP solution induces a RMSE of $9.8 \times 10^4$ atoms/g. Notice that with values ranging

between $6 \times 10^3$ and $7 \times 10^3$ atoms/g, the analytical measurement errors are 1.4 to 1.7 times smaller than the RMSE of the

MAP solution. This illustrates the effect of model errors. If the model would have been perfect, the MAP solution should have

been associated with a RMSE that is within the measurement error range of $6 \times 10^3$-$7 \times 10^3$ atoms/g (section 4.1).

Fig. 9 presents the 95 % uncertainty interval associated with model predictions. This interval brackets 5 (71% of the data) or

6 (86% of the data) observations out of 7, depending on how to consider the situation of measurement data point -03 (Table 1)

that is located at the limit of the uncertainty band. With 7 data points only it is impossible to further assess the accuracy of the

95% uncertainty band displayed in Fig. 9. Overall, it seems reasonably consistent.

**4.2.2 Comparison against the approach taken in CRONUScalc**

For CRB we sampled over an evenly-spaced 3-dimensional grid with upper and lower limits defined in section 3.3.5 and 60

grid divisions in each dimension. This leads to a total of $216 \times 10^3$ model evaluations, which is overall similar to what was

used in our proposed approach ($120 \times 10^3$). Moreover, we assumed a constant measurement error: $\sigma_{m_1}, \dots, \sigma_{m_N} = \sigma_m = 7 \times$

$10^3$ atoms/g. Using a constant rather than variable measurement error is fully justified here because (1) the analytical error range is only $6 \times 10^3 - 7 \times 10^3$ atoms/g (section 4.1), and (2) as shown in section 4.2.1 the model cannot fit the data up to the maximum measurement error of $7 \times 10^3$ atoms/g anyway.

The marginal posterior distributions of the 3 sampled parameters are presented in Fig. 10. The CRB finds similar modal or MAP values as our approach (compare Fig. 7a-c with Fig. 10a-c). The $t$ posterior distribution obtained by CRB is very close to that derived by our approach, except for a narrower peak around the MAP. For $E$ and $N_{\text{inh}}$, the posterior distribution obtained by CRB is a narrower version of that derived by our approach (compare Fig. 7a with 10a and Fig. 7c with 10c). This is caused by the use of $\sigma_m = 7 \times 10^3$ atoms/g in the likelihood function (Eq. (10)). The latter generally induces a more peaky likelihood (and consequently narrower posterior density) than our approach for which $\sigma_e$ values in the range shown in Fig. (8d) are sampled. Since similarly as for our approach the RMSE of the MAP solution derived by CRB is approximately $9.8 \times 10^4$ atoms/g, the CRB likelihood function is actually too narrow to be consistent with the achieved data misfit. This leads herein to a moderate underestimation of uncertainty. This becomes more apparent in the resulting predictive uncertainty intervals (Fig. 11). Indeed the 95% uncertainty band only brackets 4 data points out of 7, that is, 57% of the observations.

Lastly, it is worth noting that the fact that the distributions presented in Fig. 7 are less smooth than those showed in Fig. 10 is due to the different natures of grid-based sampling and MCMC. A given bin in Fig. 7 is made of 3000 samples that are drawn from the posterior pdf by the MCMC, while each bin in Fig. 10 corresponds to a single (central-bin) point of the sampled lattice.

### 4.2.3 Accounting for model errors in CRONUScalc

A simple fix for making the CRB uncertainty estimates more consistent in the presence of model errors is as follows:

I.   Identify the minimum RMSE over the sampled lattice, plug it as an estimate of $\sigma_e$ in Eq. (6) and compute the posterior density of each lattice point.

II.  Check whether the resulting MAP solution has a RMSE that is close to the fixed $\sigma_e$. These two values will obviously be equal if uniform priors are used, but may not necessarily be similar otherwise.

III. If the above is satisfied, then proceed with the inference. Otherwise, set $\sigma_e$ to the RMSE of the MAP solution, re-compute the posterior density of each lattice point and go back to II.

For the considered case study, this procedure expectedly leads to fixing $\sigma_e$ to about $9.8 \times 10^4$ atoms/g. This results into increased posterior parameter and predictive uncertainty that get relatively close to that derived by our approach in Fig. 7 and 8. Yet these uncertainty estimates remain slightly smaller than those displayed in Fig. 7 and 8. This is because rather than fixing $\sigma_e$ our approach infers its complete posterior distribution given the information content of the measurement data.

**5 Discussion of the obtained erosion rate estimate**

In Fig. 12, erosion rates for outcrops, as published by Portenga and Bierman (2011) are shown, together with the mean erosion rate obtained in the present study. The global erosion data are based on surface samples (thickness ranging between 0.5 cm and 8 cm) from a variety of bedrocks, including igneous, metamorphic and sedimentary rocks, and various climato-tectonic settings. Generally speaking, outcrop erosion rates from Portenga and Bierman (2011) tend to be lower than the 39 ± 8 m/Myr determined for the northwestern part of the Campine Plateau. Since the data set of Portenga and Bierman (2011) is entirely based on bedrock samples, the higher erosion rate of the sandy deposits can reflect differences in erosion resistance of the substrate, i.e., consolidated rock versus unconsolidated sediment. Nevertheless, in a western European context, the erosion rate that we report for the northwestern Campine Plateau seems to be fairly high for a fluvial terrace. For comparison, the Meuse younger Main Terrace (YMT) near Liège does not show any signs of post-depositional erosion following Rixhon et al. (2011). Probably, the coarse-grained and slightly consolidated nature of the Meuse gravels can be put forward as an explanation. Similarly, Dehnert et al. (2011) reported that the High Terraces of the Rhine in Germany and the Netherlands were eroded by only 1-3 m, and that the loess cover presumably protected the Rhine sands from significant erosion soon after deposition.

An alternative explanation for the relatively high erosion rate found in the current study may be the proximity of the North Sea, and low base level during glacial periods. In contrast to the Meuse and Rhine, the Scheldt basin, to which the northwestern Campine Plateau belongs today, developed in response to the sudden base level lowering as a result of the opening of the English Channel, ca. 450 ka (see Section 2). An important feature of the Scheldt basin is the Flemish Valley, a buried river system. The sudden base level lowering may have caused a regressive erosion wave penetrating into the hinterland, shaping the Flemish Valley and its eastern extension, i.e. the Nete catchment, and cause increased erosion rates in this distal part of the Scheldt basin. The posterior distribution for $t$, showing an increasing probability for $t < 0.5$ Myr, and peaking between 200 ka and 50 ka supports this hypothesis. In the case of non-stationary erosion, it remains unclear to which extent the erosion rate can be used to infer the total amount of erosion at the site. In this study, we used $E \times t$ as a joint prior for total erosion for which a lower and upper limit of 1 m and 35 m was set, and our results show an erosion estimate distribution as given in Fig. 13. The posterior for $E \times t$ is poorly resolved, which is mainly caused by the poorly resolved posterior for $t$.

Our erosion estimates for the top of the northwestern Campine Plateau asks for a revision of regional landscape evolution models. Firstly, our results suggest that the total amount of Rhine sediment would have been larger than what can be observed today in the quarries (Fig. 4); this should be taken into account when correlating Rhine sediment from the Campine Plateau with that in the Roer Valley Graben. Secondly, they indicate that post-depositional fault movement along (segments of) the Feldbiss fault as derived from the stratigraphy of Rhine sands should be considered as a minimum (Fig. 3). Thirdly, the amount of post-depositional erosion west of the plateau (Fig. 2) as can be observed from present-day altitude differences (northwestern Campine Plateau vs. Nete valley; Fig. 4b) should be regarded as a minimum erosion value.

In future work, we plan to consider more parameters within the Bayesian inversion when new and more densely sampled profiles become available. Increasing parameter dimensionality is straightforward with our MCMC sampling, but may quickly become intractable with the pure grid-based approach taken in the CRONUScalc program (Marrero et al., 2016). This limitation also holds for plain MC simulation as done by Hidy et al. (2010). Moreover, in an attempt to account for differences in geomorphological process rates in the regions of sediment provenance or non-stationary sedimentation rates, resulting in grain-size dependent $^{10}$Be concentrations, our MCMC inversion could be combined with a distributed numerical forward modelling approach instead of the currently used analytical solution.

## 6 Conclusion

We inverted within a Bayesian framework an in-situ produced $^{10}$Be concentration depth profile from the northwestern Campine plateau (NE Belgium) to infer the average long-term erosion rate, surface exposure age and the inherited $^{10}$Be concentration in the profile. Compared to the state-of-the-art in probabilistic inversion of CRN profile data, our inversion approach has two new ingredients: it (1) uses Markov chain Monte Carlo (MCMC) sampling, and (2) accounts (under certain conditions) for the contribution of model errors to posterior parameter and predictive uncertainty. We compared our approach to that taken in the CRONUScalc program for the considered case study. Both approaches are found to produce similar maximum a posteriori (MAP) values. Nevertheless, the method implemented in CRONUScalc also moderately underestimates uncertainty. A simple fix for making these uncertainty estimates more consistent in the presence of model errors is therefore proposed. For the studied fluvial terrace of the Rhine which today belongs to the Nete catchment (Scheldt basin), the derived MAP post-depositional erosion rate is ca. 39 ± 8 m/Myr (1σ). This is fairly high compared to published outcrop erosion rate data in the Meuse and Rhine catchment, and elsewhere in the world. We believe that the unconsolidated and gravel-poor nature of the studied Rhine sediment, the absence of a protecting cover (such as loess) and possibly also headward erosion in response to sudden base level lowering around 450 ka are possible explanations. Our future work will try to better resolve erosion rate together with several other uncertain parameters from MCMC inversion of dense two-nuclide concentration depth profiles.

## 7 Code availability

Python codes of the two Bayesian inversion approaches used in this study are available from https://bitbucket.org/ericlaloy/CRN_probabilistic_inversion/.

## 8 Author contributions

432 E. Laloy implemented and performed the inversions, and jointly prepared the manuscript with K. Beerten and V. Vanacker. K. Beerten designed the study and performed the field sampling. V. Vanacker prepared the field samples for CRN analyses, 434 processed the raw measurement data and supervised the presented work. B. Rogiers co-designed the study. M. Christl performed the AMS measurements at ETH-Zurich. L. Wouters supervised the work on behalf of ONDRAF/NIRAS.

## 9 Acknowledgements

This work is performed in close cooperation with, and with the financial support of NIRAS/ONDRAF, the Belgian Agency 438 for Radioactive Waste and Fissile Materials, as part of the programme on geological disposal of high-level/long-lived radioactive waste that is carried out by ONDRAF/NIRAS.

440

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

Figure Captions
Figure 1 – Location of the Campine region within Europe and the European Sand Belt.
Figure 2 – Structural map of northwestern Europe showing the Roer Valley Graben faults and the Brabant and Rhenohercynian
       (Ardennes) massifs superimposed on a Digital Terrain Model (DTM) of northwestern Europe (GTOPO30; data available from
the U.S. Geological Survey), with indication of large rivers ([http://www.eea.europa.eu/data-and-maps/data/wise-large-rivers-](http://www.eea.europa.eu/data-and-maps/data/wise-large-rivers-and-large-lakes)
       [and-large-lakes](http://www.eea.europa.eu/data-and-maps/data/wise-large-rivers-and-large-lakes)) and location of the Scheldt basin, (dashed line), the Nete catchment (dotted line) and the Flemish Valley (solid
line). The general palaeohydrography of the Meuse and Rhine between 0.5 Ma and 1.0 Ma is shown in coloured lines.
       Headward erosion as an explanation for the development of the Nete catchment is indicated with a yellow arrow.
       Figure 3 – DTM of the Campine Plateau (Digitaal Hoogtemodel Vlaanderen II, DTM, raster, 1 m) and the extent of Rhine
deposits in the study area (shading; based on Beerten (2005) and Deckers et al. (2014)).
Figure 4 – (a): Detailed DTM of the study area (Digitaal Hoogtemodel Vlaanderen II, DTM, raster, 1 m), with indication of
       the sampling location (white arrow). Note the regularly shaped sand quarries south of profile line A-A' which appear as
depressions on the DTM. (b): Topographic cross-section according to the profile line (A-A') shown in (a). The sampling
       location is schematically shown as a grey rectangle.
       Figure 5 – Photograph of the sampled profile with indication of sampling points, field codes, lithological units (A-G) and
approximate profile depth.
Figure 6 – $^{10}$Be concentration profile and results of the grain size analyses. Note that the elevated $^{10}$Be concentrations belong
       to samples that were analysed using a smaller grain size fraction than the other samples (i.e., 250-500 µm instead of 500-1000
µm). These are indicated by pale gray dots. Depth is given relative to the uppermost sample.
Figure 7 – Marginal posterior distributions of the four sampled parameters by our approach: (a) erosion rate, (b) exposure age.
       (b), (c) inherited $^{10}$Be concentration and (d) standard deviation of the residual errors. The blue bars denote the posterior pdfs
and the red lines signify the associated prior pdfs.
Figure 8 – Selected scatter plots together with iso-density contour lines for the sampled posterior parameter distribution by our
       approach. Black dots are posterior parameter sets and the density increases with the line color ranging from red (lower density)
to yellow (higher density).
Figure 9 – 95 % predictive uncertainty interval (gray area) and associated MAP prediction (black line) derived by our approach. The red crosses represent the seven measurement data points used in the inversion. Depth is given in absolute depth.

       Figure 10 – Marginal posterior distributions of the three sampled parameters by the Bayesian approach taken in the
CRONUScalc program: (a) erosion rate, (b) exposure age. (b), and (c) inherited $^{10}$Be concentration. The blue bars denote the posterior pdfs and the red lines signify the associated prior pdfs.

       Figure 11 – 95 % predictive uncertainty interval (gray area) and associated MAP prediction (black line) derived by the Bayesian
approach taken in the CRONUScalc program. The red crosses represent the seven measurement data points used in the inversion. Depth is given in absolute depth.

       Figure 12 – Frequency distribution of outcrop erosion rates published by Portenga and Bierman (2011), with indication of the
mean value obtained for the northwestern Campine Plateau (this study).
Figure 13 – Derived posterior distribution for total erosion (denudation) at the study site ($E \times t$).

Table 1 - Analytical results from the in-situ produced $^{10}$Be analysis. The depth profile is located at 50,95°N and 5,63°W at an altitude of 45 m. A SLHL production rate of 4.25 ± 0.18 atoms/g/yr was used, which represents the regionally averaged SLHL production for Europe (Martin et al. 2015). We refer to the manuscript for more information on the methodology used.

| Sample label | Sample field code | Relative depth (cm)* | Absolute depth (cm) | Quartz (g) | Be carrier (mg) | $^{10}$Be/$^9$Be (× 10$^{-12}$) | $^{10}$Be conc (× 10$^5$atoms/g qtz) |
|---|---|---|---|---|---|---|---|
| TB1204 | BE-MHR-II-00 | 0 | 45 | 34,406 | 0,208 | 0,388 ± 0,016 | 1,537 ± 0,065 |
| TB1205 | BE-MHR-II-01 | 30 | 75 | 33,535 | 0,207 | 0,329 ± 0,016 | 1,328 ± 0,070 |
| TB1206 | BE-MHR-II-02 | 50 | 95 | 33,370 | 0,207 | 0,252 ± 0,016 | 1,015 ± 0,070 |
| TB1207 | BE-MHR-II-03 | 70 | 115 | 34,467 | 0,207 | 0,318 ± 0,016 | 1,245 ± 0,065 |
| TB1940 | BE-MHR-II-04 | 110 | NA | 23,478 | 0,164 | 0,521 ± 0,019 | 2,397 ± 0,095 |
| TB1208 | BE-MHR-II-05 | 150 | 195 | 34,620 | 0,207 | 0,231 ± 0,015 | 0,898 ± 0,061 |
| TB1944 | BE-MHR-II-06 | 190 | NA | 23,486 | 0,164 | 0,709 ± 0,043 | 3,272 ± 0,204 |
| TB1209 | BE-MHR-II-07 | 230 | 275 | 34,186 | 0,207 | 0,251 ± 0,014 | 0,987 ± 0,060 |
| TB1210 | BE-MHR-II-09 | 310 | 355 | 33,663 | 0,207 | 0,229 ± 0,014 | 0,909 ± 0,060 |
| TB1211 | BE-BLANK-01 | NA | NA | 0,000 | 0,207 | 0,0011 ± 0,0006 | |
| TB1941 | BE-BLANK-02 | NA | NA | 0,000 | 0,164 | 0,0041 ± 0,0009 | |

*The relative depth $z_{fluv}$ is given as depth below the contact with the overlying aeolian sand cover.*

Table 2 – Posterior linear correlation coefficients between the 4 sampled parameters with our approach.

| Sampled parameter | $E$ | $t$ | $N_{inh}$ | $\sigma_e$ |
|---|---|---|---|---|
| $E$ | 1 | | | |
| $t$ | -0.21 | 1 | | |
| $N_{inh}$ | 0.67 | -0.56 | 1 | |
| $\sigma_e$ | -0.13 | 0.08 | -0.23 | 1 |

Figure 01

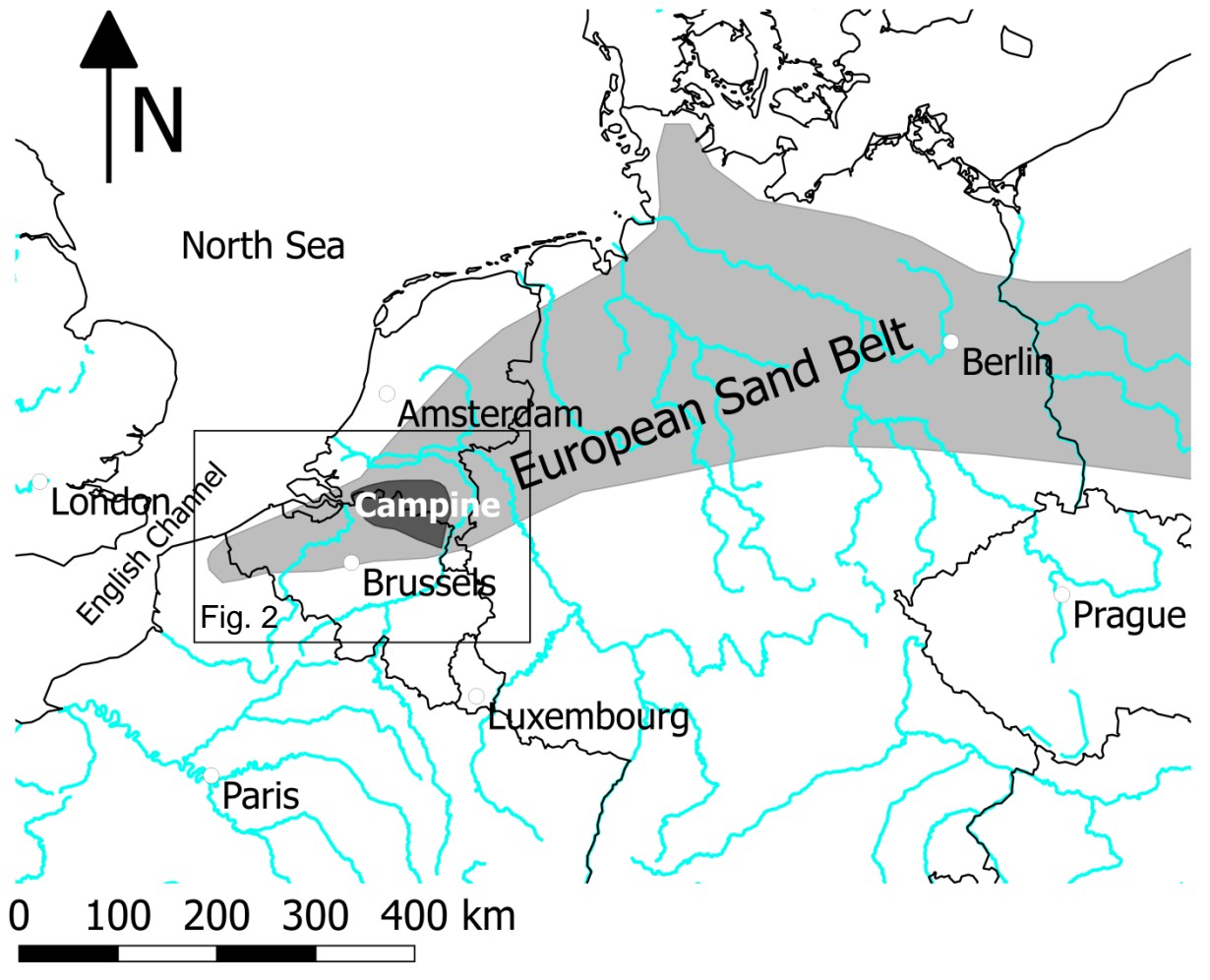

Figure 02

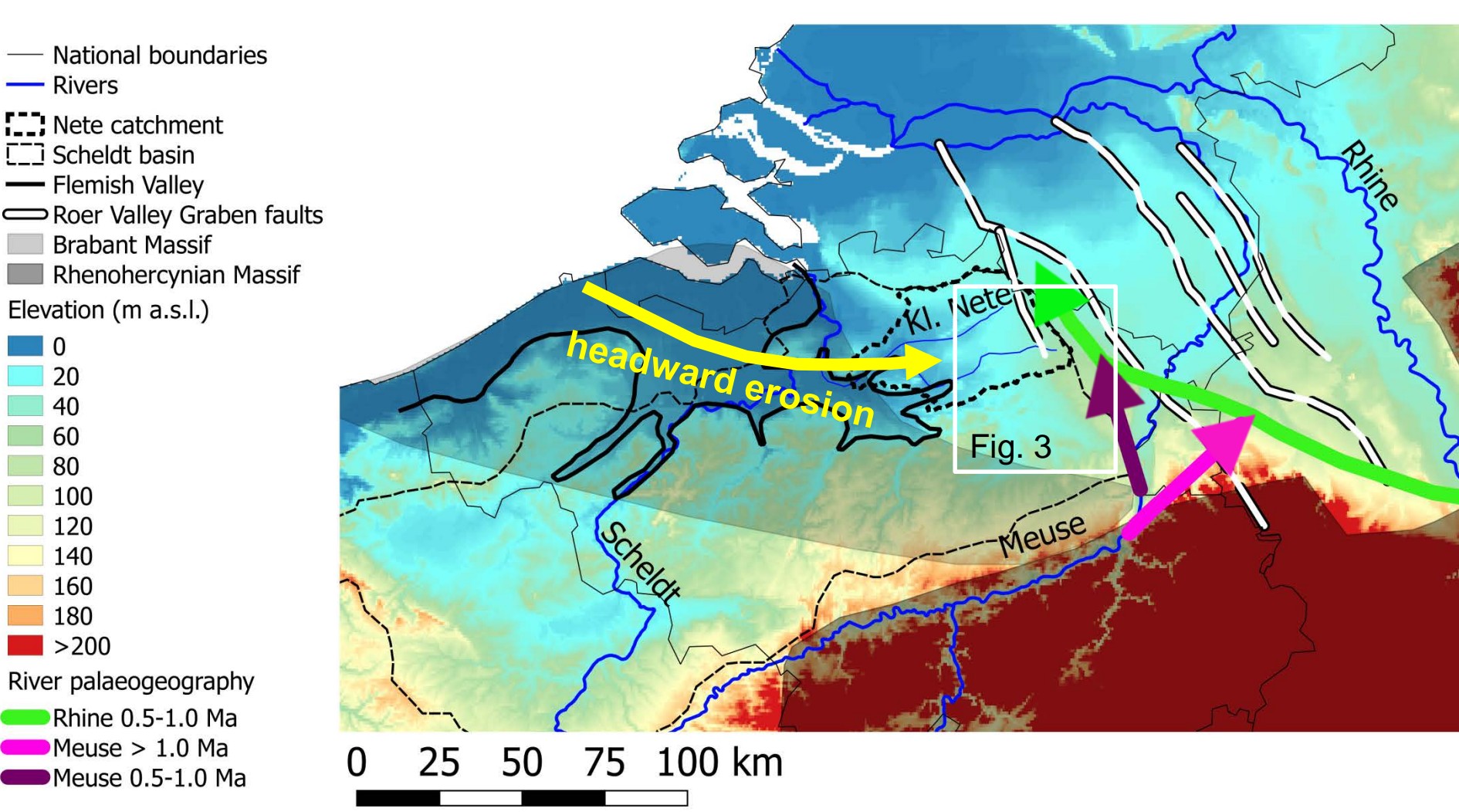

Figure 03

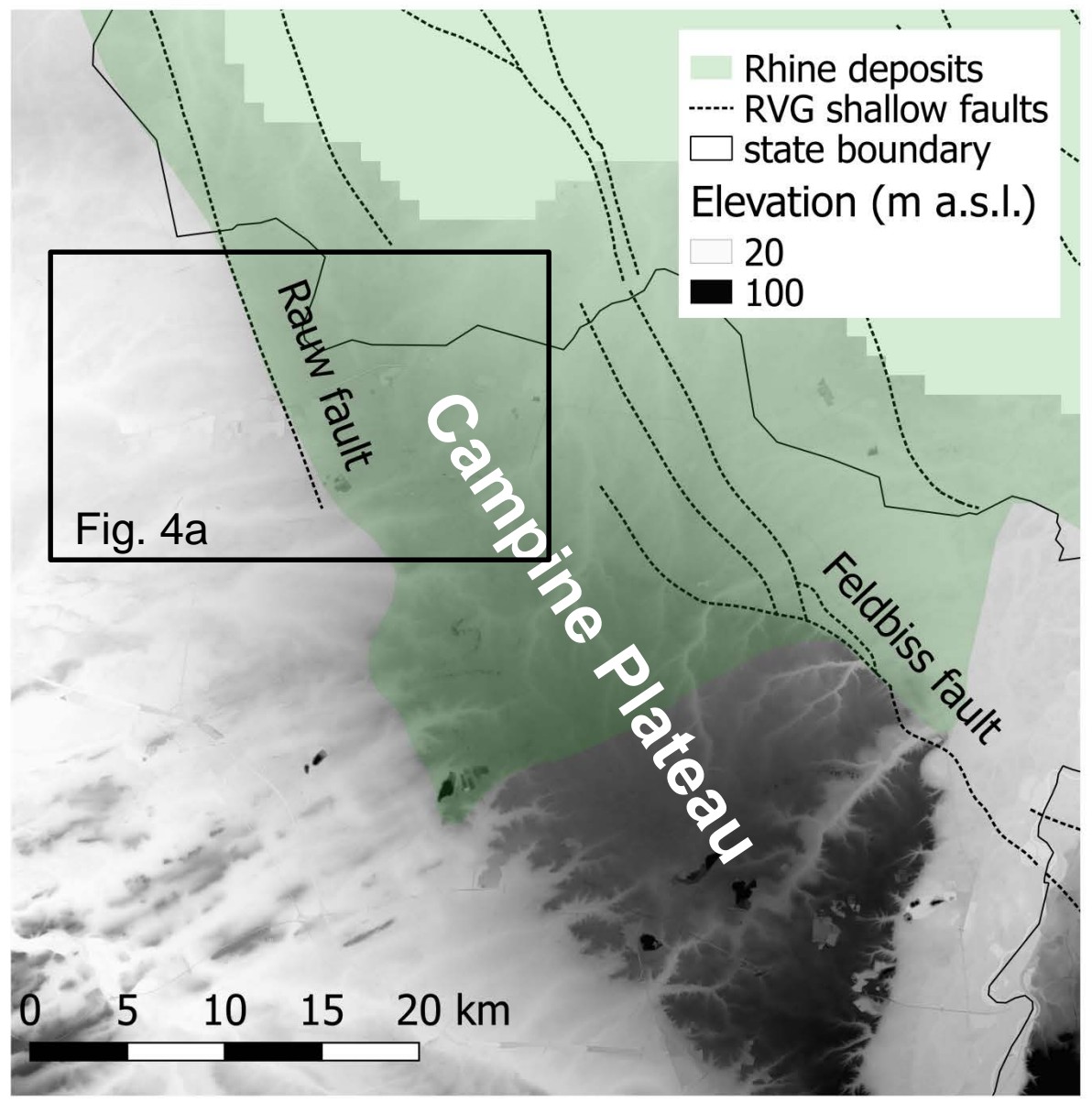

Legend:
- Rhine deposits
- RVG shallow faults
- state boundary

Elevation (m a.s.l.)
- 20
- 100

Rauw fault

Feldbiss fault

Campine Plateau

Fig. 4a

0   5   10   15   20 km

Figure 04

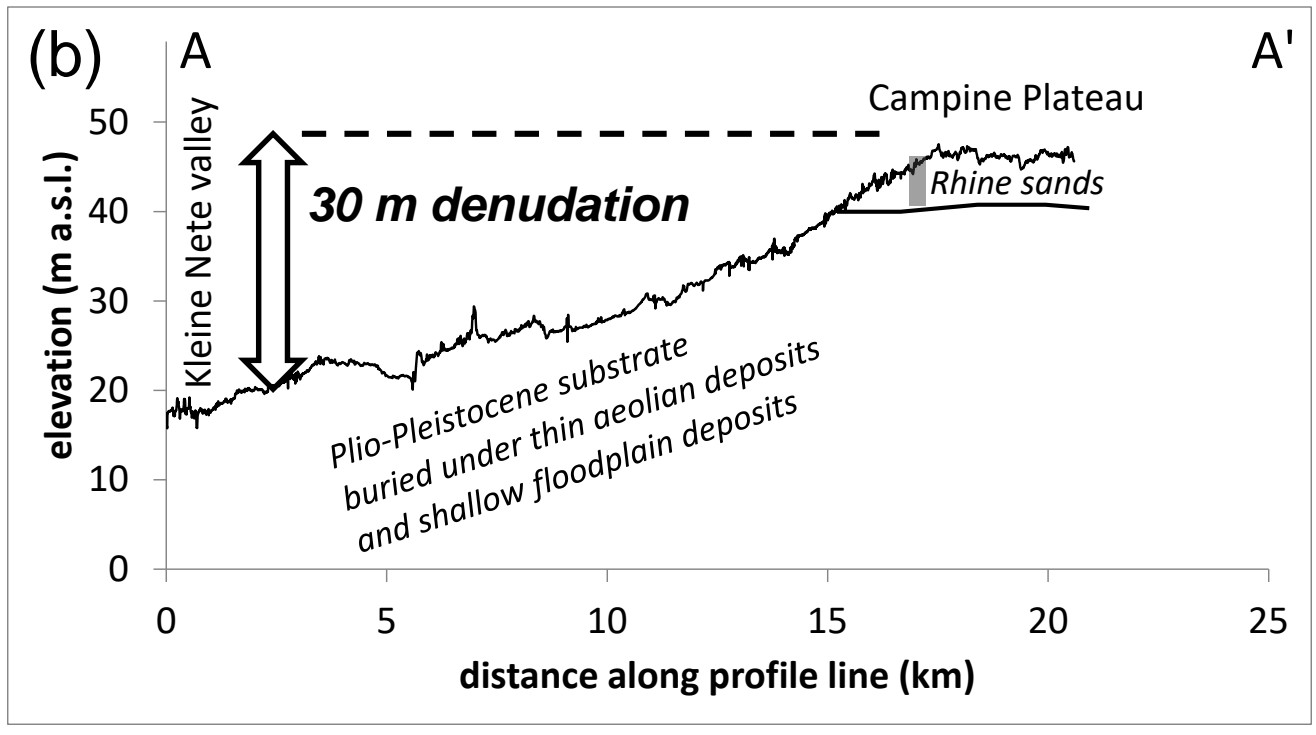

Figure 05

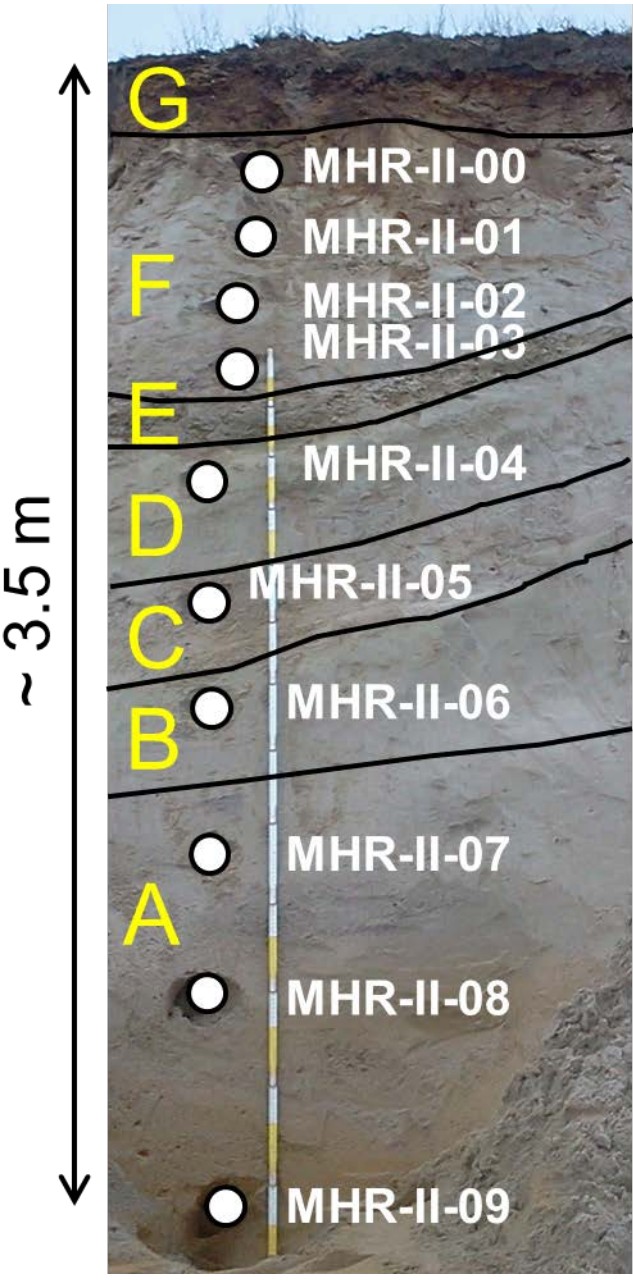

Figure 06

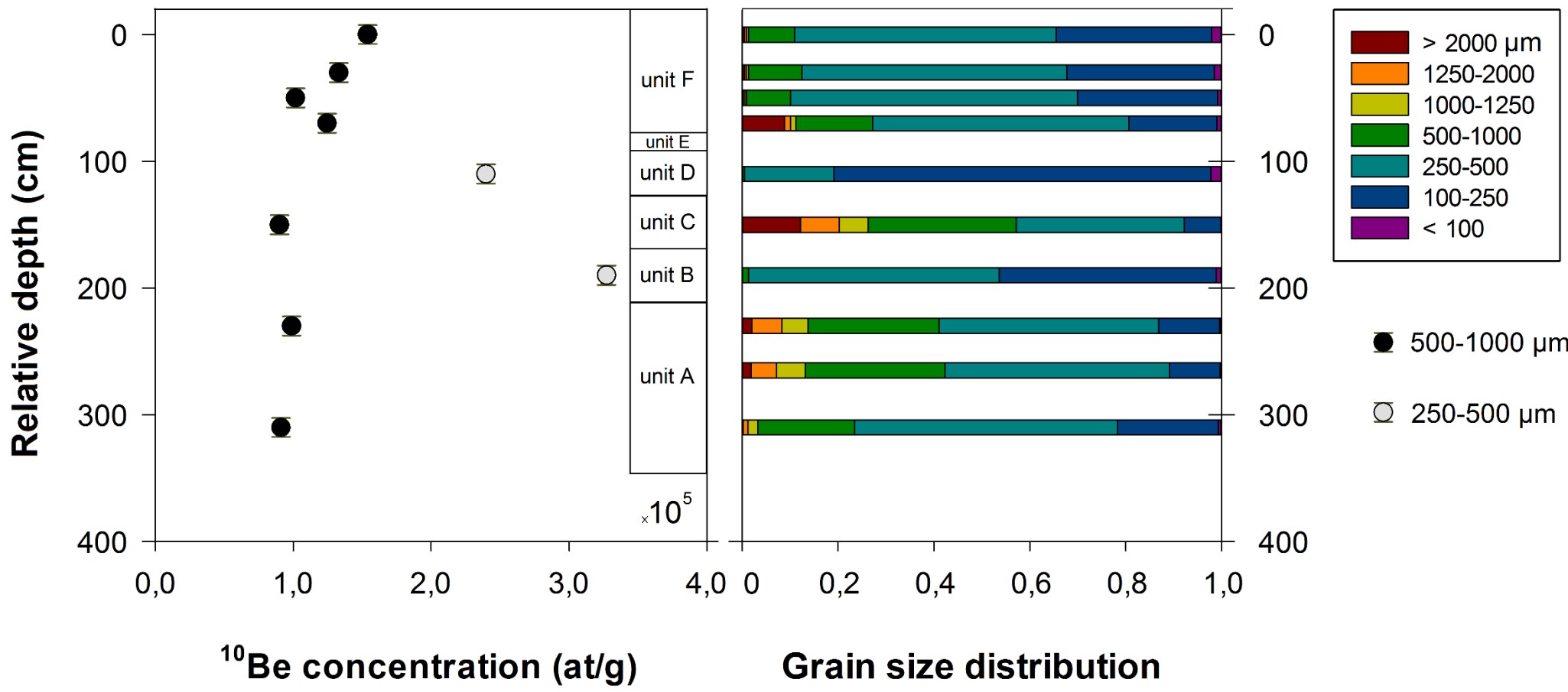

Figure 07

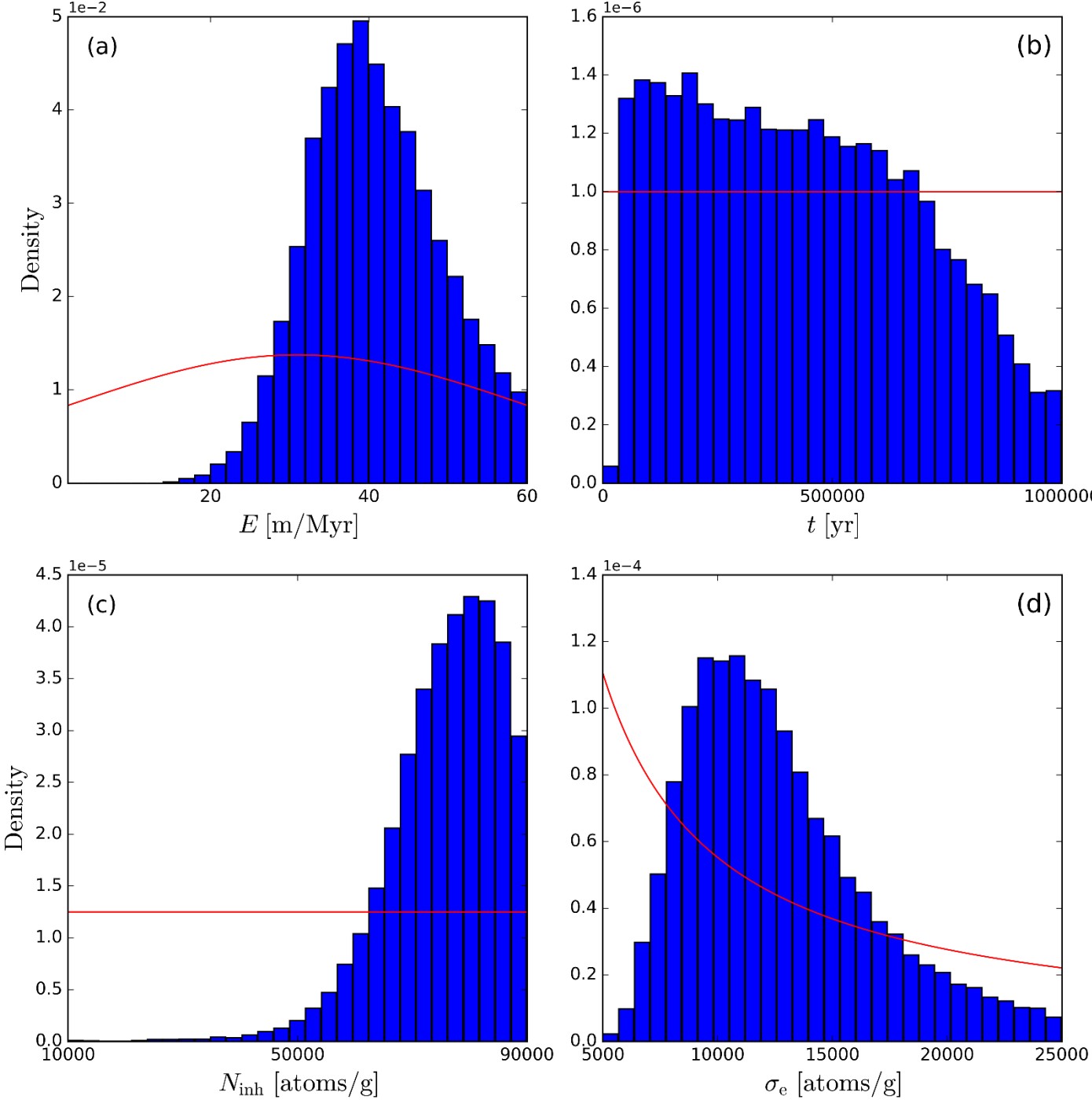

Figure 08

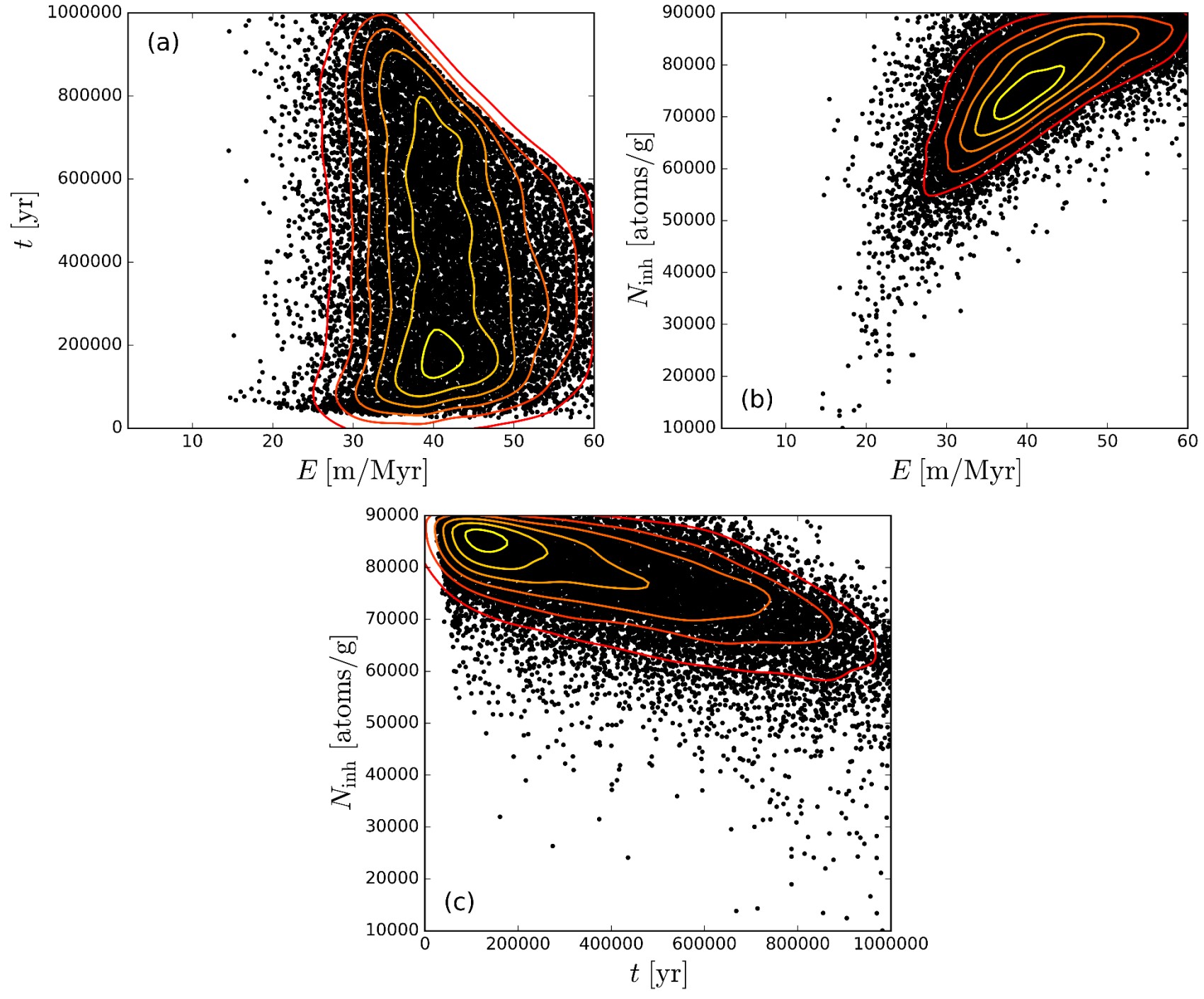

Figure 09

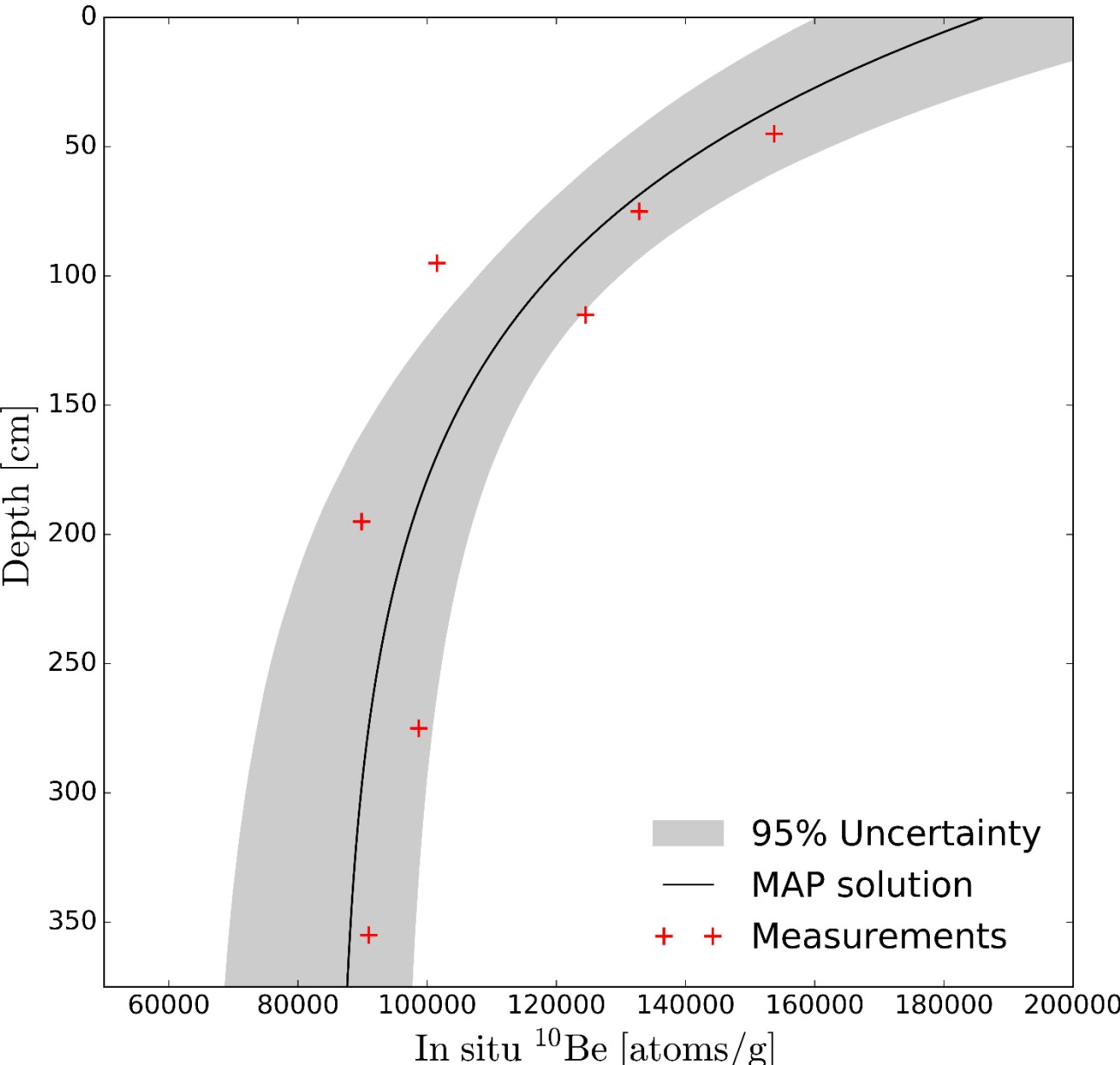

# Figure 10

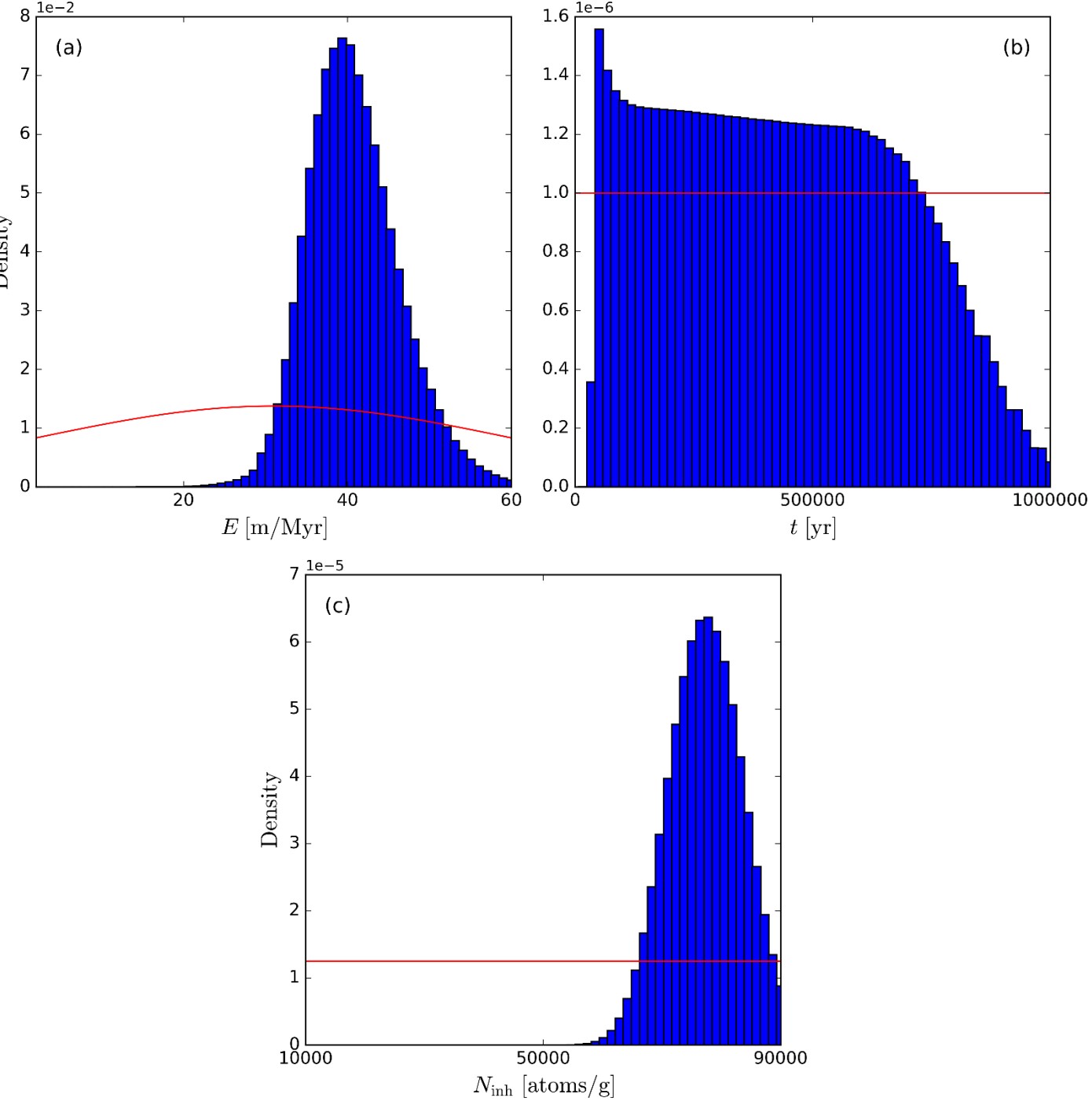

Figure 11

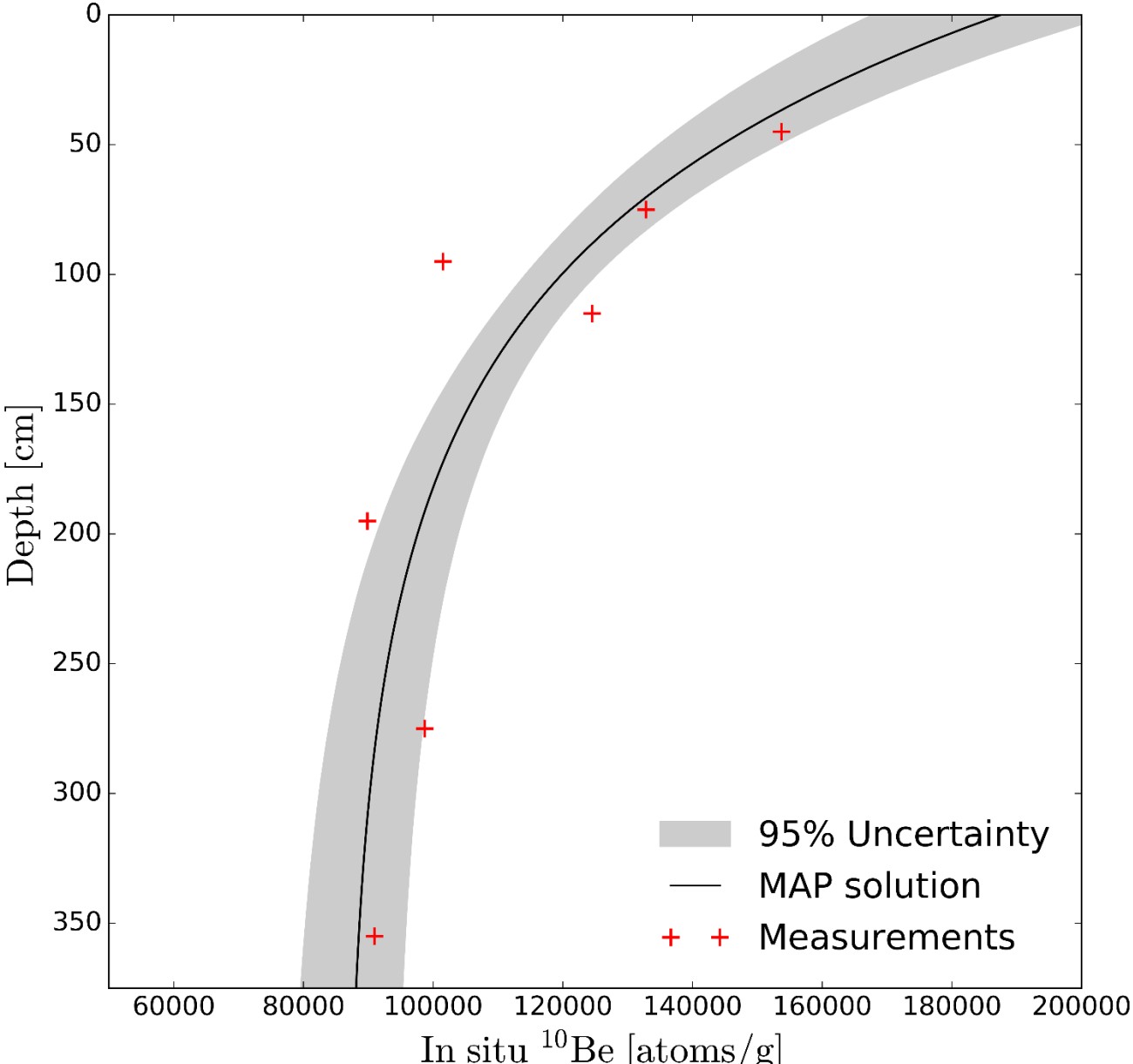

Figure 12

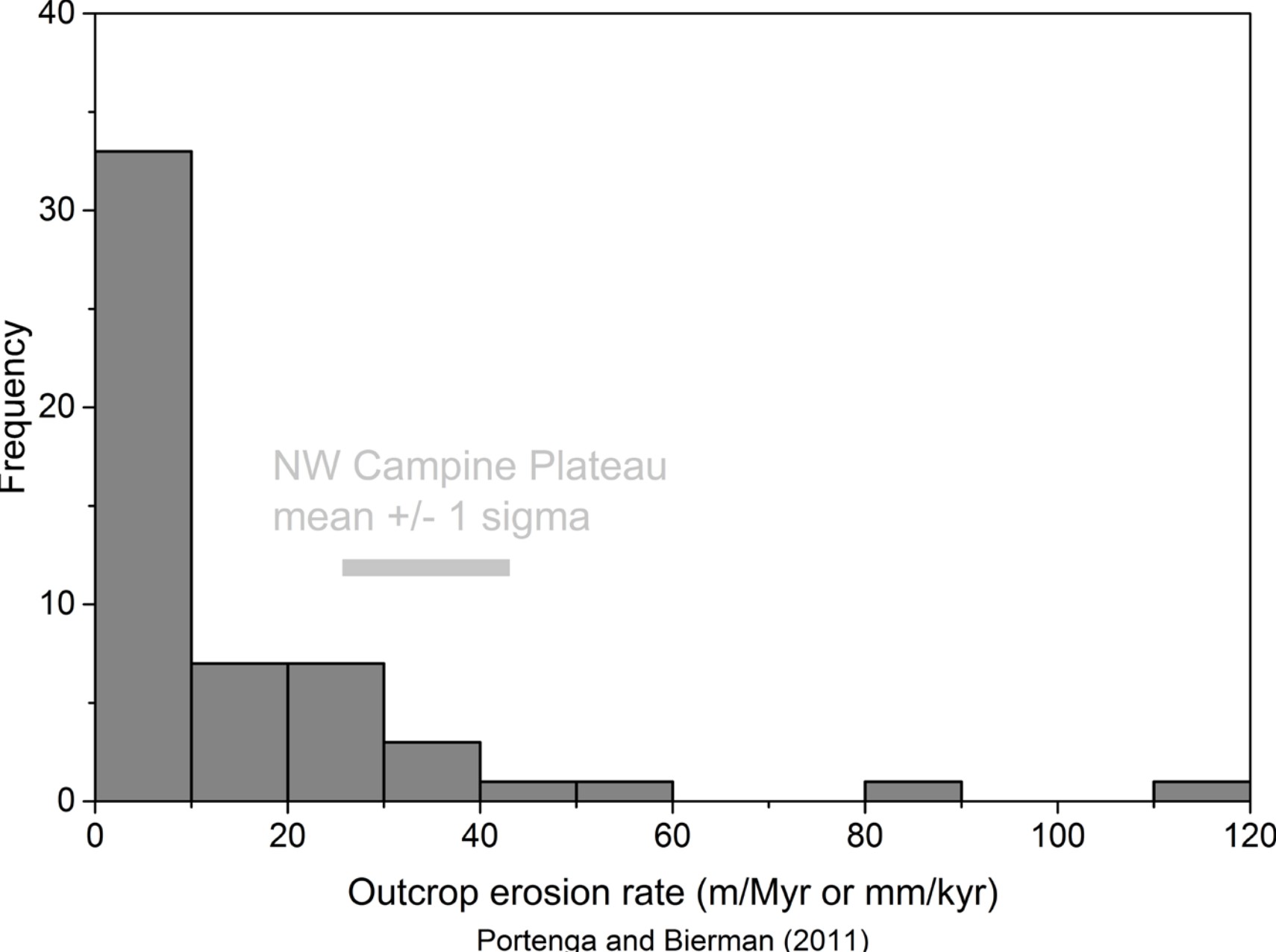

Portenga and Bierman (2011)

Figure 13

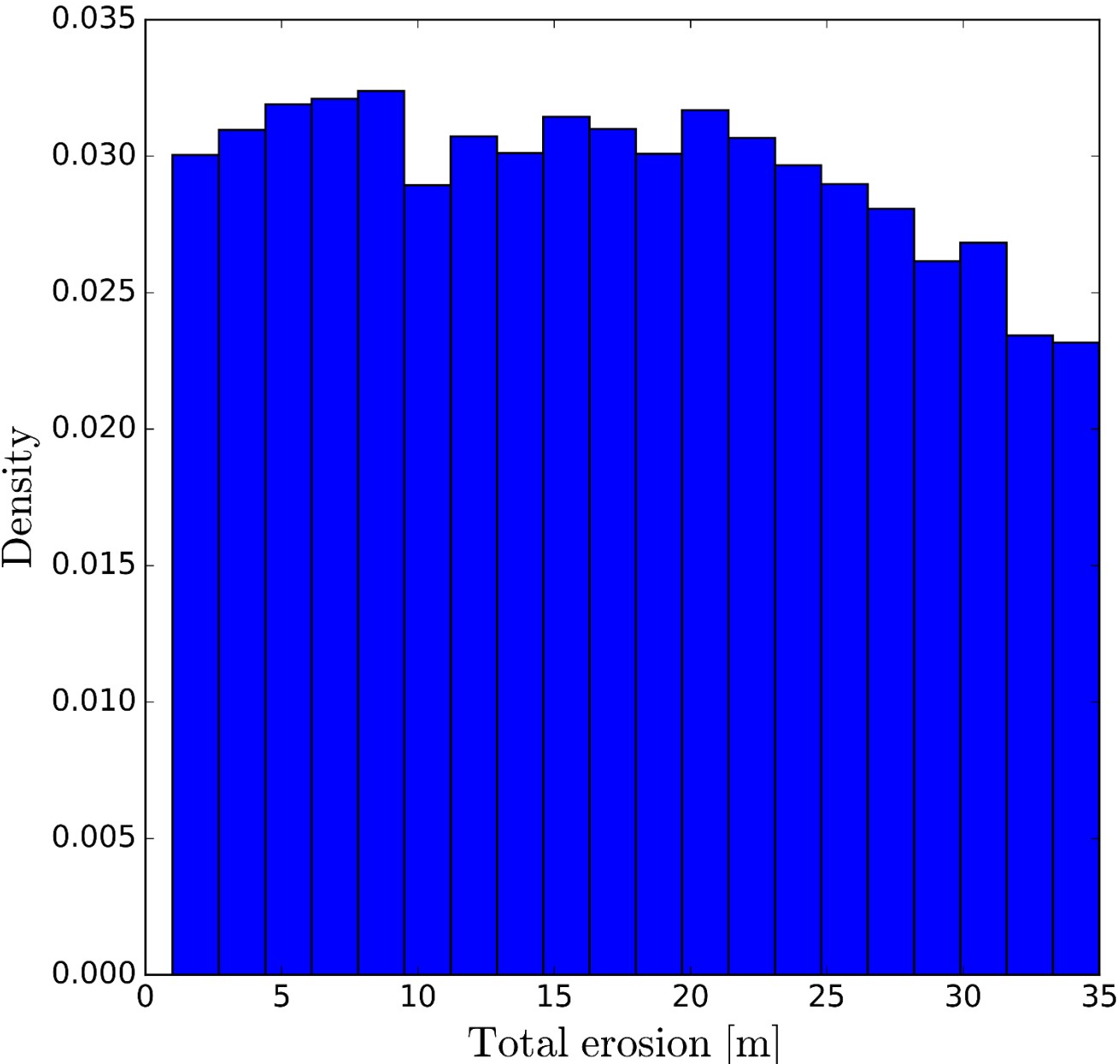