# Peer review of "Bayesian inversion of a CRN depth profile to infer Quaternary erosion of the northwestern Campine Plateau (NE Belgium)"

_Earth Surface Dynamics, 2016_

## Referee Comment (RC1) · Anonymous Referee #1 · 12 Jan 2017

The authors present a Bayesian inversion method for analyzing in situ 10Be depth profiles and use it to model data from a fluvial deposit on top of the Campine Plateau. They do a great job describing their inversion method and the framework of their Bayesian inversion approach appears sound. However, there are critical issues with how the model is applied to their dataset that need to be considered.

I think fundamentally there is some confusion over the information realistically preserved in cosmogenic depth profiles. In some cases, a measured depth profile can converge to a unique solution for both of age and erosion rate (as described by Braucher

et al. 2009). I would argue that these cases are very rare as it requires characterizing both the spallogenic and muogenic production pathways with the dataset. Most depth profiles in sediments cannot do this; typically because of large scatter in concentration with depth (which is the case for the profile presented here). Without such characterization and without other independent geologic constraint, a profile will not yield a unique solution for age and erosion rate, but it can still yield a minimum exposure age (or zero erosion age), and a maximum erosion rate for t→∞. I have included a graphic from Hidy et al. (2010) illustrating this point with an unconstrained depth profile simulation (Fig. 1):

In this graphic the red dots represent 100,000 Monte Carlo generated age-erosion rate solutions that fit within a 95% confidence of all parameter uncertainties; the blue dots represent the 500 best fits. Like many depth profiles, there is no unique age-erosion solution possible without adding some constraint to either age or erosion rate (in this case, a 30 cm net erosion cutoff was used to resolve an age). The same solution space pattern is found using the Campine Plateau dataset, but with different asymptotes for minimum age and maximum erosion rate. When the authors interpret the Campine data, they impose an age constraint of 0.5-1 Ma, which essentially restricts their solution space to the erosion asymptote and removes any variance in erosion that would be present if younger ages were permitted. It should be noted that this is perfectly OK to do if the age constraints are robust. However, depth profiles are not necessarily applicable to the timespan since deposition, and unfortunately that seems to be the case for the Campine data.

Vital to the depth profile technique is that erosion is assumed to be steady-state, and not episodic. A significant episodic erosion event (∼3-4 m) can almost completely remove a previous exposure/erosion rate signal. Sure there will be some residual signal from the deeper muogenic component, but this signal will be fairly constant with depth and probably indistinguishable from inheritance. This means that for deposits where a complicated erosion history is likely—particularly in old deposits like those

in the Campine area—a depth profile's minimum exposure age is likely not linked to depositional age, and its maximum erosion rate is likely not applicable to the time since deposition. The age-erosion rate solution space is basically only relevant to the exposure conditions following the episodic erosion. Considering this, I have the following major concerns with how the authors interpret their depth profile:

1) The authors constrain the age of their profile to between 0.5-1 Ma based on pre-existing age constraints for deposition age. However, the landform that developed following deposition could be, and likely is, significantly younger. Any age constraint applied to a depth profile must consider the time over which the surface can be assumed to be in steady-state erosion. The authors realize there is a problem in the discussion, noting that if their calculated erosion rate was correct over that timescale then it would imply an impossible amount of erosion. The authors then invoke non-steady-state erosion as the likely explanation, suggesting erosion started after 450 ka. Notably this is outside the age constraints placed on the model. What then does the erosion rate of 44 mm/ka mean? Over what timescale is this relevant? This erosion rate was obtained when assuming the surface underwent steady-state erosion for 0.5-1 Ma, but the authors argue that this couldn't have happened. If the age constraint that led to resolving the erosion rate is not viable, then the erosion rate is also not viable. The probability distribution of the erosion rate parameter needs to be re-calculated without a constraint on a lower age limit.

2) In lines 254-258 the authors reject a minimal erosion rate because it implies an apparent exposure age of 21.5 ka that conflicts with constraints on the depositional age of the deposits (0.5-1Ma). I propose that this exposure age may actually be correct, or at least closer to the age of the stable surface. What if an episode of erosion occurred at ~21.5 ka such that it wiped out earlier traces of exposure and the surface has been relatively stable ever since? How can this scenario be ruled out? Interestingly, this would suggest episodic erosion at the LGM, which seems at least plausible and shouldn't be outright dismissed. It might be useful to compare soil development at this site with data

from dated soils that may be available in the region to see if this is a scenario that can be supported or refuted.

Other minor comments:

1) Bayesian/Monte Carlo-style models have previously been applied to cosmogenic depth profiles (see version 1.2 of Hidy et al. (2010) described in Mercader et al. (2012) supplemental code; see Marrero et al. (2016), CRONUS web-based calculator). The specific MCMC algorithm employed here is new, but the authors seem to imply that this is the first depth profile model to explicitly treat parameter and model errors with Bayesian statistics, which is not the case. Work that is highly relevant to the content of this paper seems to have been ignored, starting with the original depth profile model paper of Anderson et al. (1996).

2) Thicknesses do not appear to be given for the profile samples. Or maybe I missed this? I was unable to reproduce their specific model parameter results without this information.

3) Our knowledge of production rate scaling has increased significantly over the past 5 years. The authors may want to use something more up-to-date based on all the recent calibration data. I would recommend pulling a site production rate from one of the many online calculators (e.g. CRONUS, CREP). I believe there was also an update to the Braucher muon scaling—see Braucher et al. (2011), I think.

References: Anderson et al. (1996), Explicit treatment of inheritance in dating depositional surfaces using in situ 10Be and 26Al, Geology. Braucher et al. (2009), Determination of both exposure time and denudation rate from an in situ-produced 10Be depth profile: A mathematical proof of uniqueness. Model sensitivity and applications to natural cases, Quat. Geo. Braucher et al. (2011), Production of cosmogenic radionuclides at great depth: A multi element approach, Earth and Planetary Science Letters. Hidy et al. (2010), A geologically constrained Monte Carlo approach to modeling exposure ages from profiles of cosmogenic nuclides: An example from Lees Ferry,
Arizona, Geochemistry, Geophysics, Geosystems. Marrero et al. (2016), Cosmogenic nuclide systematics and the CRONUScalc program, Quaternary Geochronology. Mercader et al. (2012), Cosmogenic nuclide age constraints on Middle Stone Age lithics from Niassa, Mozambique, Quaternary Science Reviews.
* * *

---

## Referee Comment (RC2) · Anonymous Referee #2 · 18 Jan 2017

In this manuscript, the authors apply a Bayesian inversion model on a Be-10 depth profile at a single site to investigate the erosion rate of an approx. 3000km2 area. A large part of the manuscript is dedicated to discussing in detail the geological history of the study area in the last few million years, however, almost completely ignoring at least 5 cold periods in the past 0.5 Myr. The glaciations, which most probably affected this region in many ways on multiple time-scales were taking place just north of the area implying a complex geomorphological history (deflation, permafrost, loess deposition). Additionally being situated close to the sea, the site might have experienced multiple transgressions due to isotactic rebound and sea level changes (not necessarily 0.5Myr ago or before). The study is also ignoring the fact that the Nete catchment is a highly urbanized area and not immune to recent neo-tectonism.

My major concern with the current publication is that it suggests a considerable methodological achievement with the Bayesian model and Monte Carlo type simulation based on only 7 data points (as 2 were discarded). However, using only the Be-10 concentration of the top sample in the profile and doing a quick exploratory calculation in CRONUS, using the same parameters as in the paper, yields almost the same value as the complex model, i.e. 31+/-3.11 m/Myr. The similarity in these values might be a simple coincidence, but both of them are equally valid given that there is no evidence to suggest otherwise (see Figure1)

| 10 Be results:                                   |                                                                               |                            |                                           |                                    |                            |                                                        |                               |                            |                                    |                                           |                                                 |                                    |
|-------------------------------------------------------------|-------------------------------------------------------------------------------|----------------------------|-------------------------------------------|------------------------------------|----------------------------|--------------------------------------------------------|-------------------------------|----------------------------|------------------------------------|-------------------------------------------|-------------------------------------------------|------------------------------------|
| Results not dependent on spallogenic production rate model: |                                                                               |                            |                                           |                                    |                            | Erosion rates constant production rate model:          |                               |                            |                                    |                                           |                                                 |                                    |
|                                                             |                                                                               |                            |                                           |                                    |                            | Scaling scheme for spallation: Lal(1991) / Stone(2000) |                               |                            |                                    |                                           |                                                 |                                    |
| Sample name                                                 | Shielding
factor Production rate
(muons)
(atoms/g/yr)   1.0000 0.076 |                            | roduction rate
(muons)
(atoms/g/yr) | Internal
uncertainty
(m/Myr) |                            | Erosion ra
(g/cm2/yr                                | te Erosion rate
) (m/Myr)  |                            | External
uncertainty
(m/Myr) |                                           | Production rate
(spallation)
(atoms/g/yr) |                                    |
| Belgium                                                     |                                                                               |                            | 0.43                                      |                                    | 0.00527                    | 32.92                                                  |                               | 2.                         | 2.49                               |                                           | 4.27                                            |                                    |
| Erosion rates time-varyi                                    | ng produc                                                                     | ction mo                   | dels:                                     |                                    |                            |                                                        |                               |                            |                                    |                                           |                                                 |                                    |
| Scaling scheme
for spallation:                           | Desilets and others (2003,2006)                                               |                            |                                           | Dunai
(2001)                    |                            |                                                        | Lifton and othe
(2005)     |                            | thers                              | Time-dependent
Lal (1991)/Stone (2000) |                                                 | ident
e (2000)                  |
| Sample name                                                 | Erosion
rate
(g/cm2/yr)                                                 | Erosion
rate
(m/Myr) | External
uncertainty
(m/Myr)        | Erosion
rate
(g/cm2/yr)      | Erosion
rate
(m/Myr) | External
uncertainty
(m/Myr)                     | Erosion
rate
(g/cm2/yr) | Erosion
rate
(m/Myr) | External
uncertainty
(m/Myr) | Erosion
rate
(g/cm2/yr)             | Erosion
rate
(m/Myr)                      | External
uncertainty
(m/Myr) |
| Belgium                                                     | 0.00495                                                                       | 30.92                      | 14.13                                     | 0.00491                            | 30.72                      | 3.11                                                   | 0.00501                       | 31.34                      | 3.09                               | 0.00512                                   | 32.03                                           | 2.62                               |

Therefore, I would question if the rather complex calculations presented in the paper are valid if the underlying data is scarce and might be also incorrectly presented. Note in Table 2, Page 18 10Be/9Be ratios have 4% uncertainties but 10Be concentrations have only 1% uncertainties. Obviously this is not possible, as concentration values are obtained from the ratios. This would also imply that the **Prediction uncertainty interval** section in Line 234-240 is incorrect as the authors add a max. analytical measurement error to the dataset of 2000 atoms/g, instead of ~ 20000 atoms/g. This error in data reporting also negates the conclusion of lines 274-282 *"It is interesting to note that with a range between 1.4 × 103 and 2 × 103 atoms/g, the analytical measurement errors are more than 5 times smaller than the values taken by \sigma. This nicely illustrates the effect of model errors. If the model would have been perfect, the achieved RMSE values and \sigma*

distribution should indeed have been within this measurement error range of 1.4  $\times 103$  to 2  $\times 103$  atoms/g."

I would also question the upward fining trend, as data presented in Figure 5 and Figure 6 are not entirely supporting this conclusion. Further, I would question the exclusion of two crucial points in Unit D and B, ie. MHR-II-04 and MHR-II-06 data points, which coincide with different stratigraphic layers, as these might represent erosional events or depositional events, (i.e., superimposed profiles).

The real benefit of the Bayesian approach presented here would be the ability of solving these problematic cases of depth-profiles, otherwise the top sample is enough for a rough age or erosion calculation. In summary, in my opinion, the Be-10 dataset is not suitable for the complex numerical analyses that it is subjected to. These data simply do not support the conclusions of this study.

**Minor issues**

- (1) Line 44: I think it is possible to resolve processes over the last 2Myr (See Balco et al 2013) instead of 1Myr as the authors mention. Also in terms of general cosmogenic Be-10 literature it might be more suitable to quote Dunai 2010 or Gosse and Phillips 2001 rather than Hancock 1999 and Heine 2009.
- (2) Line 128: It will not change too much on the results probably, but muon attenuation lengths are different (see Braucher et al 2011), and SLHL production rate closer to 4 at/g/y (see Borchers et al 2015).
- (3) Lines 164-186: Section 3.3 (Bayesian inversion) is word by word the same as in Minet et al 2015.
- (4) Line 262: "-0.13" is not a statistically significant correlation
- (5) Figure 6: Note that unit layering conventionally is starting from top as "A" and progressing downwards, not the other way.
- (6) Presentation of figures might require revisiting.

One of the motivations (including funding of this project) behind this study was to understand the implications of long-term landscape evolution of this area on radioactive waste management. Unfortunately this study failed to address this aspect.

---

## Editor Comment (EC1) · J.K. Willenbring (Editor) · 6 Feb 2017

Dear Dr Beerten and co-authors, Your paper now has two thorough reviews. Both are somewhat encouraging but with major reservations on how this 'technique' paper is a step forward from the existing technique. As such, I strongly encourage you to submit a revised version of your paper, as usual fully taking into account and responding to the reviewers' comments. I would appreciate a comment on some misunderstandings that came up regarding how the model differs from previous models. There are a couple of methods for constructing erosion/ages from depth profiles available. It would be nice

to see a comparison of these methods and to figure out the real differences including uncertainties between them. Are you able/willing to upload the (modified?) source code as a part of the manuscript or on github so that reviewers (and eventually readers) have access to it? Thank you for submitting your work to Earth Surface Dynamics. Best wishes, Jane

—————————————————————

---

## Author Comment (AC1)

We would like to thank anonymous referee #1 for the constructive and thoughtful comments and suggestions.

Below, we respond to the suggestions of Reviewer#1 (and have copied - where relevant – the remarks in green).

**MAJOR COMMENTS**

**Reviewer#1:** I think fundamentally there is some confusion over the information realistically preserved in cosmogenic depth profiles. In some cases, a measured depth profile can converge to a unique solution for both of age and erosion rate (as described by Braucher et al. 2009). I would argue that these cases are very rare as it requires characterizing both the spallogenic and muogenic production pathways with the dataset.

**Reply:** We fully agree that one should carefully consider which information can realistically be preserved in CRN depth profiles. In our opinion, Bayesian inference is especially powerful when it comes to determine how well the model parameters can be resolved by the available data. Using Bayesian techniques, we can explore the information content that is hidden in the data, and get an uncertainty on the model estimates. Our Bayesian inverse methodology is a statistically-sound tool for quantifying to what extent the model parameters are constrained by the available measurement data. The Bayesian approach considers the model parameters to be random variables having a joint posterior probability density function (pdf). Once determined, this posterior distribution encodes all the necessary information about the parameters such as degree of (non-) uniqueness, standard deviations, correlations and dependencies (given the chosen prior distribution and likelihood function). We will better highlight the advantages of Bayesian inference in the revision.

**Reviewer#1:** Without such characterization and without other independent geologic constraint, a profile will not yield a unique solution for age and erosion rate, but it can still yield a minimum exposure age (or zero erosion age), and a maximum erosion rate for t→∞.

**Reply:** To come back to our presented results, we thus consider that they rigorously represent the information content of the investigated profile. The reviewer points out that a single depth profile usually cannot simultaneously resolve t (exposure age) and E (erosion rate). We would like to stress that this is exactly what our results are showing. With a marginal posterior distribution identical to its marginal prior distribution, the exposure age is not at all resolved (see Figures 7a and 8a). In contrast, we demonstrate that the erosion rate can be resolved, yet with a relatively large uncertainty that we quantify rigorously (see Figures 7b and 8abc, and lines 265-267).

**Reviewer#1:** When the authors interpret the Campine data, they impose an age constraint of 0.5-1 Ma, which essentially restricts their solution space to the erosion asymptote and removes any variance in erosion that would be present if younger ages were permitted. It should be noted that this is perfectly OK to do if the age constraints are robust. However, depth profiles are not necessarily applicable to the timespan since deposition, and unfortunately that seems to be the case for the Campine data…. The authors constrain the age of their profile to between 0.5-1 Ma based on preexisting age constraints for deposition age. However, the landform that developed following deposition could be, and likely is, significantly younger. Any age constraint applied to a depth profile must consider the time over which the surface can be assumed to be in steady-state erosion. … If the

age constraint that led to resolving the erosion rate is not viable, then the erosion rate is also not viable. The probability distribution of the erosion rate parameter needs to be re-calculated without a constraint on a lower age limit.

**Reply:** We fully agree with reviewer #1, and will re-run our Bayesian inversion using a wider prior distribution for the exposure age, and without putting a constraint on the lower age limit. The text, tables and figures will be revised accordingly, as this might modify the outcome of the Bayesian model.

The constraint for exposure age of 0.5 to 1 Myrs that was used in the manuscript, came from our initial assumption that the depth profile is only marginally affected by post-depositional erosion. While this assumption might now seem at odds with the current results, there is a general consensus which explains the Campine Plateau as a classical case of relief inversion, with coarse-grained fluvial deposits (gravel, sandy gravel and gravelly sand) on top protecting it from significant erosion (Paulissen, 1983; Paulissen, 1997). Our depth profile is sampled at the crest of the Campine Plateau, i.e. at the top of the geomorphic surface. Our CRN results question the consensus on the stability of the Campine Plateau: when assuming near-zero denudation rates, the apparent exposure age should be congruent with the minimum depositional age of the Rhine sands in this location that is estimated to be ca. 0.5 Ma (absolute lower age estimate, see text). The Quaternary geology of Rhine and Meuse deposits, the entire history of the Meuse terrace staircase and the evolution of the Roer Valley Graben all point to a late Early to early Middle Pleistocene age for the fluvial deposits (see text for references).

We agree that we should extent the prior distribution for t down to t=0, to capture the possibility of post-depositional erosion at the site.

**MINOR COMMENTS**

We will try to address the additional minor comments of the reviewer as follows:

**Reviewer#1:** Bayesian/Monte Carlo-style models have previously been applied to cosmogenic depth profiles (see version 1.2 of Hidy et al. (2010) described in Mercader et al. (2012) supplemental code; see Marrero et al. (2016), CRONUS web-based calculator).

**Reply:** We will better acknowledge previous literature on Monte-Carlo simulation and uncertainty estimation for CRN applications. That stated, to the best of our knowledge our work is the first to apply Bayesian uncertainty quantification to CRN modeling. Indeed, the plain Monte Carlo procedure used by Hidy et al. (2010) is not Bayesian. Hidy and coworkers do not consider Bayes theorem and from a statistical point of view, their uncertainty estimates have a different meaning than ours. We believe that our Bayesian approach is both more sound and more robust. This will be clarified in the revision.

**Reviewer#1:** Thicknesses do not appear to be given for the profile samples

**Reply :** The sample thickness is 10 cm and we will properly address this issue in the revised text.

**Reviewer#1:** Our knowledge of production rate scaling has increased significantly over the past 5 years. The authors may want to use something more up-to-date based on all the recent calibration data

**Reply:** As written in lines 129-131 of our submitted manuscript, we used the same CRN production rates as the ones used by Rixhon et al. (2011). This choice was made for the sake of comparison –as in the study by Rixhon and coworkers a Middle Pleistocene Meuse terrace around Liège was investigated. However, we tend to agree now with the fact that we should use state-of-the-art knowledge on scaling and production rates of CRN's. We will use the updated data suggested by the reviewer when re-running the Bayesian model.

**REFERENCES**

Hidy, A. J., Gosse J. C., Pederson J. L., Mattern J. P., and Finkel R. C., 2010. A geologically constrained Monte Carlo approach to modeling exposure ages from profiles of cosmogenic nuclides: An example from Lees Ferry, Arizona, Geochem. Geophys. Geosyst., 11, Q0AA10, doi:10.1029/2010GC003084.

Paulissen, E., 1983. Les nappes alluviales et les failles Quaternaires du Plateau de Campine. In: Robaszynski F, Dupuis C (eds) Guides Géologiques Régionaux – Belgique, Masson, Paris, pp. 167-170.

Paulissen, E., 1997. Quaternary morphotectonics in the Belgian part of the Roer Graben. Aardkundige Mededelingen 8, 131-134.

Rixhon, G., Braucher, R., Bourlès, D., Siame, L., Bovy, B., Demoulin, A., 2011. Quaternary river incision in NE Ardennes (Belgium)-Insights from 10Be/26Al dating of river terraces, Quaternary Geochronology, 6 (2), 273-284.

---

## Author Comment (AC2)

We thank referee #2 for having taken the time and effort to read the paper in great detail, and provide us detailed suggestions. We address the most important reviewer's concerns below, and refer to the reviewer's text (green) when relevant.

As a preamble, we would like to state that there seems to be a misunderstanding regarding the main objective of the paper, which is (see lines 70-75 and copied below) to perform a Bayesian inference of a long-term erosion rate from a sampled CRN depth profile. The method is then illustrated using a 10Be concentration depth profile from NE Belgium. In the paper, we do not suggest that the results from the depth profile are representative for the entire Campine area, nor that we aim to determine the impact of landscape evolution in NE Belgium on potential nuclear waste disposal. The latter is not within the scope of this research paper, and we believe that the reviewer's comments related to nuclear waste disposal are therefore not relevant.

> Lines 70-75 of the discussion paper read as follows:
> *The overall objective of this study is to constrain within a Bayesian framework the rate and amount of post-depositional denudation of the headwaters of the Nete catchment. The latter is part of the larger Scheldt basin and is used herein to test and demonstrate the application of Bayesian inversion to CRN concentration vs. depth profiles. The Nete catchment is an interesting test case because the upstream areas of the catchment are located at the northwestern edge of the Campine Plateau, which is covered by coarse gravelly sand from the Early-Middle Pleistocene Rhine and thus constitutes a fluvial terrace from which the depositional age nor the exposure age of the sands is well constrained (Beerten et al., in press).*

**MAJOR COMMENTS**

**Reviewer#2.**

A large part of the manuscript is dedicated to discussing in detail the geological history of the study area in the last few million years, however, almost completely ignoring at least 5 cold periods in the past 0.5 Myr. The glaciations, which most probably affected this region in many ways on multiple time-scales were taking place just north of the area implying a complex geomorphological history (deflation, permafrost, loess deposition). Additionally, being situated close to the sea, the site might have experienced multiple transgressions due to isotactic rebound and sea level changes (not necessarily 0.5Myr ago or before). The study is also ignoring the fact that the Nete catchment is a highly urbanized area and not immune to recent neo-tectonism.

**Reply:** We agree with reviewer #2 that the long-term topographic evolution of lowland Europe is affected by periglacial processes during glaciations, landscape instability during glacial-interglacial-glacial transitions, uplift and (absolute) base level lowering, and will take this along in our revision of the manuscript. However, we do not agree with the suggestion that "glaciations…, loess deposition, … and multiple transgressions" might also have affected geomorphological processes in the area. Based on previous geomorphological and geological studies in NE Belgium, we can exclude that the northern part of the Campine area in Belgium was affected by an ice sheet during any of the time periods considered in the paper. Also, there are no indications that the area would have been covered with loess (as the site is located in the "European Sand Belt", see introduction and Figure 1). Next, as stated in lines 77-79, the area has not been experiencing marine conditions since the start of the Pleistocene, even though it is situated close to the sea as the reviewer correctly observes. Finally, it is correct that large parts of the Campine area are urbanized, but we believe that this is not relevant for inferring long-term geomorphological processes when sampling locations are adequately selected.

Based on the results from one depth profile, we cannot make conclusive statements on the long-term erosion rates of NE Belgium; but can formulate some hypotheses for further testing. We will clarify this nuance in our revised manuscript. We observe – for example – that the erosion rate at the sampling location is relatively large compared to global compilations of outcrop erosion rates. This result in itself is novel and diverges from current theories on landscape evolution of NE Belgium that suggested very low erosion rates for the Campine Plateau. In the paper, we suggest that erosion is related to the erosivity of the substrate and potential base level lowering in the North Sea.

**Reviewer#2.**

My major concern with the current publication is that it suggests a considerable methodological achievement with the Bayesian model and Monte Carlo type simulation based on only 7 data points (as 2 were discarded). However, using only the Be-10 concentration of the top sample in the profile and doing a quick exploratory calculation in CRONUS, using the same parameters as in the paper, yields almost the same value as the complex model, i.e. 31+/-3.11 m/Myr. The similarity in these values might be a simple coincidence, but both of them are equally valid given that there is no evidence to suggest otherwise (see Figure1)

**Reply:** We are a bit puzzled by the reviewer's statement that *"my major concern with the current publication is that it suggests a considerable methodological achievement with the Bayesian model and Monte Carlo type simulation based on only 7 data points (as 2 were discarded)"*. To the best of our knowledge, this is the first application of the Bayesian framework to CRN depth profile analysis. As detailed in our reply to the comments of reviewer #1, we argue that Bayesian inference is a very powerful tool to rigorously evaluate how well the inferred parameters are resolved by the available measurement data. We refer to our reply to reviewer #1 for more technical details.

The reviewer suggests that CRN depth profiles are not necessary to obtain relevant information on erosion rates and states that *"using only the Be-10 concentration of the top sample in the profile and doing a quick exploratory calculation in CRONUS, using the same parameters as in the paper, yields almost the same value as the complex model, i.e. 31+/-3.11 m/Myr."* This comparison seems flawed to us, as it ignores the impact of pre-deposition inheritance on the total concentration of cosmogenic radionuclides. In alluvial landforms, such as the Campine Plateau, pre-deposition inheritance cannot be neglected (see e.g. Anderson et al., 1996; Braucher et al., 2009; Vassallo et al., 2011; Hedrick et al., 2013). Hence, the importance of CRN depth profiles for determining the exposure age and denudation rates of the sampling site on the Campine Plateau.

Furthermore, we noted that reviewer #2 derived a denudation rate of 31 m/Myr, that is actually different and smaller than the mode of our marginal posterior distribution (the maximum a-posterior (MAP) or most probable value) which is about 44 m/Myr. Also, we believe that the reviewer's uncertainty estimate of +/- 3.11 m/Myr (one sigma) is unrealistically small. Applying a Bayesian inversion framework to the 7 measurement data points, we obtain a standard deviation of 9 m/Myr (see lines 266-267 and Figure 7b). We therefore argue that our estimate and the reviewer's one are not equally valid, and consider our estimate to be much more accurate.

**Reviewer#2:** Note in Table 2, Page 18 10Be/9Be ratios have 4% uncertainties but 10Be concentrations have only 1% uncertainties. Obviously this is not possible, as concentration values are obtained from the ratios.

**Reply:** With respect to measurement uncertainty: there was indeed a mistake in the [10]Be concentration errors as they were reported in Table 1, and we thank the reviewer for pointing this out. The corrected values for the [10]Be concentration errors are in the range 6500 – 7000 atoms/g.

**Reviewer#2:** This would also imply that the prediction uncertainty interval section in Line 234-240 is incorrect as the authors add a max. analytical measurement error to the dataset of 2000 atoms/g, instead of ~ 20000 atoms/g. This error in data reporting also negates the conclusion of lines 274-282.

**Reply:** After correction, the [10]Be concentration errors are max. 7000 atoms/g (and not 20,000 as suggested by the reviewer). Since we infer $\sigma$ jointly with the other parameters, this does not affect the derived posterior parameter distribution. However, it will modify the total predictive uncertainty (light gray band in Figure 9) in the sense of a reduction as the $\sigma_m$ value in the equation of line 239 becomes 7000 atoms/g. We would also like to stress that this still illustrates the effect of model errors, although its effect is less strong. Our best fitting error is about 10000 atoms/g and the 3000 atoms/g difference can thus be attributed to model errors. We will correct Table 1, Figure 9 and the associated discussion in the revision.

**Reviewer#2:** I would also question the upward fining trend, as data presented in Figure 5 and Figure 6 are not entirely supporting this conclusion. Further, I would question the exclusion of two crucial points in Unit D and B, ie. MHR-II-04 and MHR-II-06 data points, which coincide with different stratigraphic layers, as these might represent erosional events or depositional events, (i.e., superimposed profiles).

**Reply:** Concerning the sample selection: out of the 9 samples, we selected 7 samples for the Bayesian model. Two samples were excluded from the analysis, as their CRN concentration was measured on the grain size fraction 250-500µm (instead of 500-1000 µm). Our results show that the CRN concentrations of these two samples are higher than expected when one assumes a monotonic decline of CRN concentration with depth. Currently, it is not possible to know the exact reason for the observed difference in CRN concentration, as this can either reflect a grain-size dependent CRN concentration (see e.g. Schaller et al., 2001 for European river sediments; Carretier et al., 2015), or non-stationary sedimentation. Further analyses are necessary to make a conclusive statement.

**Reviewer#2 concludes that :** *"The real benefit of the Bayesian approach presented here would be the ability of solving these problematic cases of depth-profiles, otherwise the top sample is enough for a rough age or erosion calculation. In summary, in my opinion, the Be-10 dataset is not suitable for the complex numerical analyses that it is subjected to. These data simply do not support the conclusions of this study."*

**Reply:** As pointed out above, we disagree with this statement. For the reasons mentioned above, one cannot infer a site-specific denudation rate and exposure age from one sample taken at the top of an alluvial deposit. In our opinion, the strength of the Bayesian approach is that it provides a robust way for exploring the data information content that is hidden in CRN depth profiles. Hence, we consider that our paper provides a fair illustration of the benefits of the Bayesian approach.

**MINOR COMMENTS**

Line 44: I think it is possible to resolve processes over the last 2Myr (See Balco et al 2013) instead of 1Myr as the authors mention. Also in terms of general cosmogenic Be-10 literature it might be more suitable to quote Dunai 2010 or Gosse and Phillips 2001 rather than Hancock 1999 and Heine 2009.

**Reply:** Correct, we will reformulate this sentence in the revised document, and insert the references suggested.

Line 128: It will not change too much on the results probably, but muon attenuation lengths are different (see Braucher et al 2011), and SLHL production rate closer to 4 at/g/y (see Borchers et al 2015).

**Reply:** (See also reply to reviewer #1): As written in lines 129-130 of our submitted manuscript, we used the same CRN production rates as the ones used by Rixhon et al. (2011). This choice was made for the sake of comparison – as in the study by Rixhon and coworkers a Middle Pleistocene Meuse terrace close to Liège (Belgium) was investigated. However, we now agree with the suggestion that we should use state-of-the-art knowledge on scaling and production rates of CRN's. We will use the updated data suggested by the reviewer and rerun the Bayesian model.

Lines 164-186: Section 3.3 (Bayesian inversion) is word by word the same as in Minet et al 2015.

**Reply:** The second author, Eric Laloy, has written the section on Bayesian statistics in the present manuscript and is the co-author who has done the Bayesian analysis in the paper by Minet et al. (2015), of which he is second author as well.

Line 262: "-0.13" is not a statistically significant correlation

**Reply:** When writing "significant" we did not mean "statistically significant", which only makes sense in the context ofd statistical hypothesis testing. To avoid any confusion, we will replace "significant" by "substantial" or similar in the revised text.

Figure 6: Note that unit layering conventionally is starting from top as "A" and progressing downwards, not the other way.

**Reply:** We believe that this is a matter of taste – examples of the ordering we use can be found in the literature.

Presentation of figures might require revisiting

**Reply:** We will check every figure for the revision and provide improvement if deemed necessary.

**Reviewer#2 ends the review stating that :** "One of the motivations (including funding of this project) behind this study was to understand the implications of long-term landscape evolution of this area on radioactive waste management. Unfortunately this study failed to address this aspect"

**Reply:** We believe that this statement is out of scope. As stated earlier, the aim of the paper is to constrain a site-specific denudation rate using a Bayesian framework. It is nowhere written in our manuscript that our paper contributes to understand the implications of long-term landscape

evolution on potential radioactive waste disposal. Research on long-term landscape evolution and denudation rates can be informative for land management, and waste disposal research.

**REFERENCES**

Anderson R.S., Repka J.L., Dick G.S.? 1996. Explicit treatment of inheritance in dating depositional surfaces using in situ 10Be and 26Al, Geology, 24, 47–51.

Braucher R., Del Castillo P., Siame L., Hidy A.J., and Bourlés D.L., 2009. Determination of both exposure time and denudation rate from in situ-produced 10Be depth profile: a mathematical proof of uniqueness. Model sensitivity and applications to natural cases, Quaternary Geochronology, 4, 56–67

Carretier, S., Regard, V., Vassallo, R., Aguilar, G., Martinod, J., Riquelme, R., Christophoul, F., Charrier, R., Gayer, E., Farías, M., Audin, L., Lagane, C., 2015. Differences in 10Be concentrations between river sand, gravel and pebbles along the western side of the central Andes, Quaternary Geochronology, 27, 33-51.

Hedrick K., Owen L.A., Rockwell T.K., Meigs A., Costa C., Caffee M.W., Masana E., Ahumada E., 2013. Timing and nature of alluvial fan and strath terrace formation in the Eastern Precordillera of Argentina, Quaternary Science Reviews, 80, 143-168.

Rixhon, G., Braucher, R., Bourlès, D., Siame, L., Bovy, B., Demoulin, A., 2011. Quaternary river incision in NE Ardennes (Belgium)-Insights from 10Be/26Al dating of river terraces, Quaternary Geochronology, 6 (2), 273-284.

Schaller M., von Blanckenburg F., Hovius N., Kubik P.W., 2001. Large-scale erosion rates from in situ-produced cosmogenic nuclides in European river sediments, Earth Planet. Sci. Lett., 188, 441–458.

Vassallo R., Ritz J.-F., and Carretier S., 2011. Control of geomorphic processes on 10Be concentrations in individual clasts: Complexity of the exposure history in Gobi-Altay range (Mongolia), Geomorphology, 135, 35-47.